# BIDORA: BI-LEVEL OPTIMIZATION-BASED WEIGHT-DECOMPOSED LOW-RANK ADAPTATION

## ABSTRACT

Parameter-efficient fine-tuning (PEFT) of large language models (LLMs) has gained considerable attention as a flexible and efficient way of adapting LLMs to downstream tasks. Among these methods, weighted decomposed low-rank adaptation (DoRA) has emerged as a promising approach. DoRA bridges the gap between low-rank adaptation (LoRA) and full fine-tuning (FT) by decomposing the weight matrices into magnitude and direction components, thereby maintaining learning behavior similar to FT. Although DoRA shows encouraging performance, it introduces additional parameters compared to LoRA, which potentially increases the risk of overfitting. Moreover, optimizing magnitude and direction simultaneously leads to a coupled gradient updating pattern for both components, limiting its learning capacity. To overcome these limitations, we propose BiDoRA, a bi-level optimization-based PEFT method. In BiDoRA, the direction and magnitude components are optimized on two distinct datasets at different optimization levels, mitigating the risk of overfitting. Additionally, the asynchronous optimization of the two components promotes their decoupling, allowing for more flexible gradient updates suitable for various downstream tasks. Evaluation of BiDoRA on fourteen datasets spanning natural language understanding, natural language generation, and token classification reveals that it significantly outperforms DoRA and other PEFT methods. The superior performance of BiDoRA underscores its effectiveness. The code for BiDoRA is available at https://anonymous.4open.science/r/BiDoRA-5D31.

## 1 INTRODUCTION

Large language models (LLMs) (Radford et al., 2019; Brown et al., 2020) have achieved state-of-the-art results across a broad range of NLP tasks, from natural language understanding (NLU) (Wang et al., 2019) to natural language generation (NLG) (Novikova et al., 2017). Although LLMs are pre-trained on extensive datasets in the general domain, additional methods are required to adapt them to specific downstream datasets for better performance. For example, full fine-tuning (FT) involves updating all pre-trained parameters on downstream datasets (Qiu et al., 2020), which typically yields superior results. However, as LLMs continue to scale, FT incurs significant computational costs and increases the risk of overfitting during the fine-tuning process (Karimi Mahabadi et al., 2021). Alternatively, in-context learning adapts LLMs to new tasks by incorporating a few data examples directly into the prompt, eliminating the need for parameter updates (Brown et al., 2020). While this approach substantially reduces computational costs and mitigates the risk of overfitting, it is constrained by the use of only a limited number of examples from the downstream data, often leading to suboptimal performance.

To address the limitations above, parameter-efficient fine-tuning (PEFT) methods (Houlsby et al., 2019; Hu et al., 2021) have been introduced as a promising solution. PEFT approaches update only a subset of the pre-trained parameters, achieving performance comparable to FT while requiring significantly fewer computational resources. Among these methods, low-rank adaptation (LoRA, Hu et al. (2021)) has gained popularity due to its simplicity and effectiveness. LoRA attaches low-rank matrices to the pre-trained model weights, updating only these matrices during fine-tuning without changing the model architecture. However, LoRA exhibits a parameter updating pattern distinct from FT, which may limit its learning capacity (Liu et al., 2024). Specifically, when decomposing the parameter updates of both methods into magnitude and direction components, the

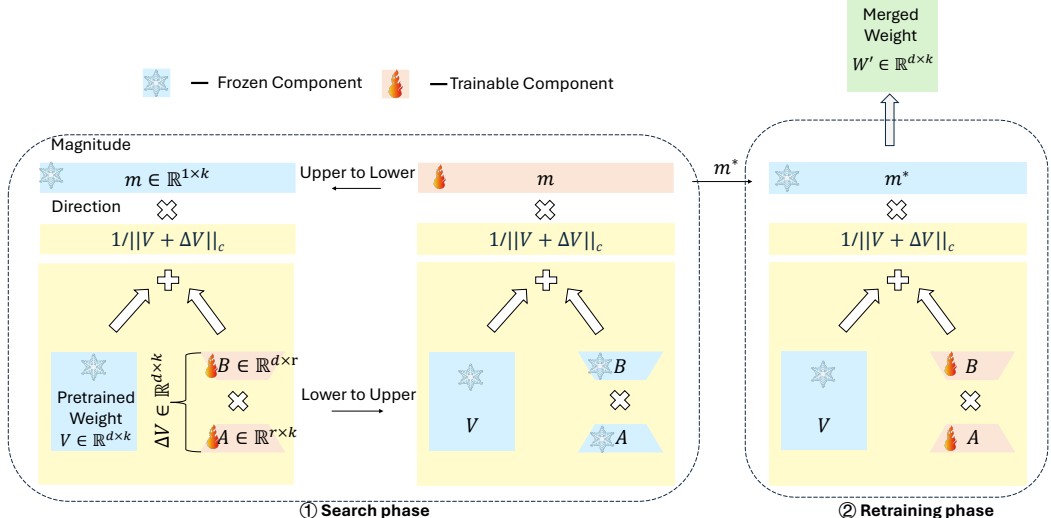

Figure 1: **An overview of BiDoRA.** BiDoRA performs PEFT using a bi-level optimization framework. At the lower level, BiDoRA learns the incremental direction component $\Delta V$ of the update matrices using the training split of the downstream dataset. At the upper level, BiDoRA optimizes the magnitude component $m$ with optimized $\Delta V$ from the lower level, using the validation split of the dataset. After determining the optimal magnitude, the direction component undergoes further fine-tuning on a combined set of both training and validation splits to maximize overall performance.

correlation between these components tends to be positive in LoRA, whereas it is negative in FT. To bridge this gap, weight-decomposed low-rank adaptation (DoRA, Liu et al. (2024)) introduces an explicit reparameterization of the pre-trained weights matrix as the product of magnitude and direction components. The model can be trained end-to-end in the same way, with direction components parameterized with incremental low-rank matrices. This approach enables DoRA to share similar learning patterns with FT, thereby outperforming LoRA in multiple tasks. Nonetheless, DoRA introduces additional parameters compared to LoRA, which can exacerbate overfitting issues, particularly when adapting to smaller downstream datasets (See Figure 2). Furthermore, in DoRA, the magnitude and incremental direction components are optimized concurrently, leading to a highly constrained updating pattern that may overlook the diverse learning patterns required for different downstream tasks.

To address the challenges above, we propose BiDoRA, a bi-level optimization-based weight-decomposed low-rank adaptation method for parameter-efficient fine-tuning of LLMs. BiDoRA mitigates overfitting and facilitates a flexible weight updating pattern by separately optimizing the magnitude and incremental direction components on different splits of the downstream datasets using distinct optimization steps. BiDoRA is based on a bi-level optimization framework: At the lower level, the low-rank incremental direction component is updated using the training split of the downstream dataset, while the magnitude component remains fixed. At the upper level, the magnitude component is updated by minimizing the loss on the validation split via hypergradient descent. These two optimization steps are performed iteratively until convergence. Figure 1 provides an overview of BiDoRA.

In BiDoRA, the two distinct components are trained on separate splits of the dataset, which effectively reduces the risk of overfitting. A similar strategy is utilized in the well-established practice of differentiable neural architecture search (DARTS, Liu et al. (2018)), where architecture and model parameters are learned using different splits of the training dataset. Since the architecture selection module introduces additional parameters beyond the base model, directly optimizing this overly complex supernetwork can result in severe overfitting. Similarly, by treating the magnitude component as the architecture and the incremental direction component as the model in neural architecture search, training these components on separate datasets helps reduce overfitting. As shown in Figure 2, BiDoRA demonstrates better resistance to overfitting compared to DoRA, given the smaller performance gap between the training set and test set. Furthermore, the asynchronous gradient up-

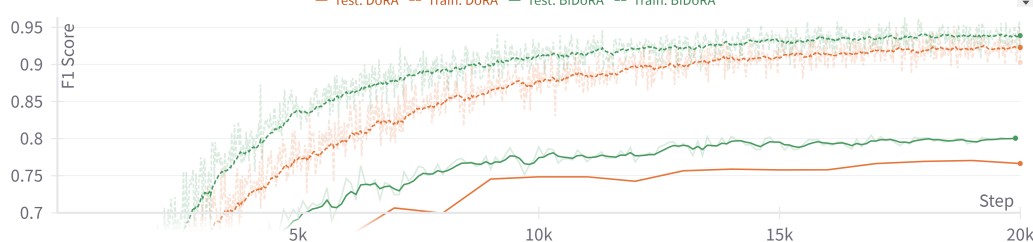

Figure 2: Training and test accuracy versus global training steps on the ModHayes split of the Reuters21578 dataset when fine-tuning a RoBERTa-base model using DoRA and BiDoRA. The training and test curves for DoRA show a larger gap compared to BiDoRA, highlighting the effectiveness of our method in reducing overfitting.

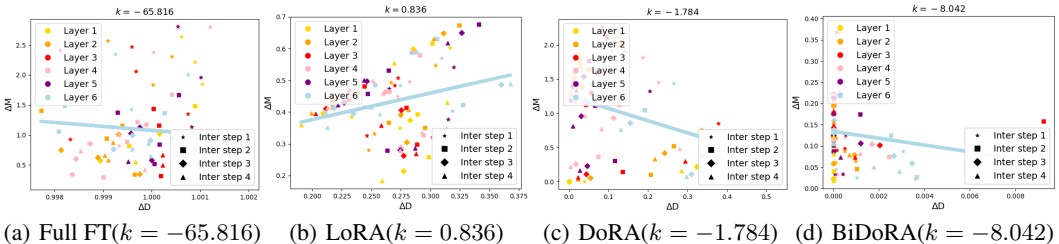

(a) Full FT($k = -65.816$)    (b) LoRA($k = 0.836$)    (c) DoRA($k = -1.784$)    (d) BiDoRA($k = -8.042$)

Figure 3: Magnitude and direction updates for (a) FT, (b) LoRA, (c) DoRA, and (d) BiDoRA of the query matrices across different layers and intermediate steps after fine-tuning the GPT2 model on the E2E dataset, where $k$ denotes the correlation value. Different markers represent matrices from different training steps, with each color corresponding to a specific layer.

date steps at the two optimization levels in BiDoRA facilitate better decoupling of the magnitude component from the incremental direction component, leading to a more flexible update pattern that closely resembles FT. As illustrated in Figure 3, the updates of the magnitude and direction components across different layers using BiDoRA have a correlation value that is closest to that of FT, highlighting its superior learning capability compared to both DoRA and LoRA.

Our work makes the following key contributions:

- We propose BiDoRA, a novel parameter-efficient fine-tuning method based on bi-level optimization. In contrast to DoRA, which trains the magnitude and incremental direction components on a single dataset, BiDoRA optimizes these components on different splits of a downstream dataset through distinct optimization steps.

- Our strategy effectively mitigates the risk of overfitting and results in a parameter update pattern that more closely resembles full fine-tuning.

- Extensive experiments on fourteen datasets highlight the superior performance of BiDoRA. BiDoRA consistently surpasses several baseline methods, including LoRA and DoRA, across tasks such as text classification, language generation, and token classification.

## 2 RELATED WORK

### 2.1 PARAMETER EFFICIENT FINE-TUNING METHODS

Parameter-efficient fine-tuning (PEFT) methods aim to reduce the high costs associated with fully fine-tuning large-scale models by updating only a relatively small subset of pre-trained parameters, rather than the entire model, to adapt to downstream tasks. Existing PEFT methods can be mainly categorized into three types. The first category, known as adapter-based methods, injects additional trainable modules into the original frozen backbone. For instance, Houlsby et al. (2019) suggests adding linear modules in sequence to existing layers, while He et al. (2021) proposes integrating these modules in parallel with the original layers to enhance performance. The second category is prompt tuning methods, which add extra soft tokens (prompts) to the initial input. During the fine-tuning stage, only these trainable soft tokens are updated, as demonstrated in works such as Lester et al. (2021) and Razdaibiedina et al. (2023). Unfortunately, the first two categories lead to increased

inference latency compared to fully fine-tuned models. The third category is low-rank adaptation methods, pioneered by the foundational work LoRA (Hu et al., 2021). These methods attach low-rank matrices to pre-trained weights and use only these matrices for weight updates during fine-tuning. Since low-rank updates can be merged with pre-trained weights before inference, low-rank adaptation-based PEFT methods do not increase inference time. Following LoRA, Zhang et al. (2023) applies SVD decomposition to low-rank matrices and prunes less significant singular values for more efficient updates. Zhang et al. (2024b) uses meta-learning to search for the optimal rank of LoRA matrices, further improving its performance on downstream tasks. Most recently, Liu et al. (2024) uses weight decomposition analysis to reveal that LoRA exhibits a distinct weight updating pattern compared to FT, which may constrain its learning capacity. Therefore, DoRA (Liu et al., 2024) was then proposed to bridge the gap between LoRA and FT. DoRA decomposes the pre-trained weights into two components—magnitude and direction—and fine-tunes both, which results in a more closely aligned updating pattern compared to FT.

## 2.2 BI-LEVEL OPTIMIZATION

Bi-level optimization (BLO) has been widely applied in various machine learning tasks, including meta-learning (Finn et al., 2017; Rajeswaran et al., 2019), neural architecture search (Liu et al., 2018; Zhang et al., 2021), and hyperparameter optimization (Lorraine et al., 2020; Franceschi et al., 2017). Despite its wide usage, solving BLO problems can be challenging due to the inherent nature of nested optimization problems. Several algorithms have been proposed to address this challenge, including zeroth-order methods such as Bayesian optimization (Cui & Bai, 2019) and first-order algorithms based on hypergradients (Pearlmutter & Siskind, 2008; Lorraine et al., 2020). Among these approaches, gradient-based BLO has received significant attention because it can scale to high-dimensional problems with a large number of trainable parameters. In this work, we extend the application scenarios of gradient-based BLO to develop a robust and effective parameter-efficient fine-tuning method for pre-trained models.

## 3 PRELIMINARY: WEIGHT-DECOMPOSED LOW-RANK ADAPTATION

LoRA (Hu et al., 2021) involves attaching the product of two low-rank matrices to the pre-trained weights and fine-tuning these low-rank matrices on downstream datasets with the pre-trained weights frozen. It is based on the assumption that parameter updates made during fine-tuning exhibit a low intrinsic rank. Formally, given a pre-trained weight matrix $W_0 \in \mathbb{R}^{d \times k}$, LoRA attaches a low-rank update matrix $\Delta W \in \mathbb{R}^{d \times k}$ to the pre-trained weight. This update matrix can be further decomposed as $\Delta W = BA$, where $B \in \mathbb{R}^{d \times r}$ and $A \in \mathbb{R}^{r \times k}$ are two low-rank matrices, with $r \ll \min(d, k)$. Consequently, the weight matrix $W'$ after LoRA fine-tuning can be represented as follows:

$$W' = W_0 + \Delta W = W_0 + BA \tag{1}$$

In this setup, the pre-trained weight matrix $W_0$ remains fixed during the fine-tuning process, while the LoRA matrix $\Delta W$ is updated. However, after performing weight decomposition on fine-tuned weight matrices, it was found that LoRA and full fine-tuning exhibit different learning patterns (Liu et al., 2024). To bridge this discrepancy, weight-decomposed low-rank adaptation (DoRA, Liu et al. (2024)) further reparameterizes the weight matrices by explicitly decomposing them into learnable magnitude and direction components. Formally, DoRA performs adaption as follows:

$$W' = m \frac{V + \Delta V}{\|V + \Delta V\|_c} = m \frac{W_0 + BA}{\|W_0 + BA\|_c} \tag{2}$$

where the incremental direction component $\Delta V$ is parameterized as a product of two learnable low-rank matrices, $B$ and $A$, while the magnitude component $m \in \mathbb{R}^{1 \times k}$ is a learnable vector. Here, $\| \cdot \|_c$ represents the vector-wise norm of a matrix computed across each column. In DoRA, both components are optimized concurrently on a single downstream dataset.

## 4 METHODS

### 4.1 OVERVIEW OF BIDORA

Our method, BiDoRA, optimizes the trainable parameters in DoRA layers by solving a bi-level optimization problem. Let $\mathcal{M} = \{m_1, m_2, \ldots, m_n\}$ denote the set of magnitude components for all $n$ DoRA modules, and $\mathcal{V} = \{\Delta V_1, \Delta V_2, \ldots, \Delta V_n\}$ denote the set of corresponding incremental direction components. Specifically, we first learn the incremental direction components $\mathcal{V}^*(\mathcal{M})$ on the training split of the downstream dataset $\mathcal{D}_{tr}$ at the lower level. The magnitude component $\mathcal{M}$ is tentatively fixed at this level, thus the resulting optimal incremental direction component $\mathcal{V}^*(\mathcal{M})$ is a function of $\mathcal{M}$. At the upper level, we determine the optimal magnitude component $\mathcal{M}^*$ by optimizing the loss on a validation split $\mathcal{D}_{val}$. In practice, $\mathcal{D}_{tr}$ and $\mathcal{D}_{val}$ are typically created by splitting the original training set without using additional data. This bi-level optimization problem is solved using an efficient gradient-based algorithm, where parameters in two levels are optimized iteratively until convergence. Related convergence analyses of this type of gradient-based bi-level optimization algorithms can be found in Pedregosa (2016), Rajeswaran et al. (2019), and references therein. The generalization analysis has also been studied in Bao et al. (2021).

### 4.2 ORTHOGONAL REGULARIZATION

The orthogonality of neural network weights has been identified as a beneficial property (Bansal et al., 2018) and can effectively mitigate the overfitting issue (Balestriero & richard baraniuk, 2018). Therefore, we define a Gram regularization loss (Xie et al., 2017) for the direction component:

$$\mathcal{R}(\mathcal{V}) = \sum_{k=1}^{n} \left\| (V_k + \Delta V_k)^\top (V_k + \Delta V_k) - I \right\|_F^2 \tag{3}$$

where $I$ is the identity matrix and $\| \cdot \|_F$ denotes the Frobenius norm. Intuitively, $\mathcal{R}(\mathcal{V})$ encourages each column of the direction matrix, representing a specific direction, to be orthogonal to one another. Since each column has already been normalized (equivalent to projected to the unit sphere), this also prompts each column to be far away from the other, thereby reducing the redundancy of parameters.

### 4.3 A BI-LEVEL OPTIMIZATION FRAMEWORK

**Lower Level** At the lower level, we train the low-rank incremental direction component $\mathcal{V}$ by minimizing a loss $\mathcal{L}_{tr}$ defined on the training set $\mathcal{D}_{tr}$. The overall training objective at this level is $\mathcal{L}_{tr}(\mathcal{V}, \mathcal{M}) = C(\mathcal{V}, \mathcal{M}; \mathcal{D}_{tr}) + \gamma \mathcal{R}(\mathcal{V})$. Here, $C$ represents the fine-tuning loss, given the low-rank incremental direction component $\mathcal{V}$, the magnitude component $\mathcal{M}$, and the training split $\mathcal{D}_{tr}$ of the downstream dataset. $\mathcal{R}(\mathcal{V})$ is the orthogonal regularizer defined in Eq. (3), with $\gamma$ as a trade-off hyperparameter. In this level, we only update $\mathcal{V}$ while keeping $\mathcal{M}$ fixed, resulting in the following optimization problem:

$$\mathcal{V}^*(\mathcal{M}) = \arg \min_{\mathcal{V}} \ \mathcal{L}_{tr}(\mathcal{V}, \mathcal{M}) \tag{4}$$

where $\mathcal{V}^*(\mathcal{M})$ denotes the optimal solution for $\mathcal{V}$ in this problem, which is a function of $\mathcal{M}$.

**Upper Level** At the upper level, we validate the previously fixed magnitudes $\mathcal{M}$ on the validation set $\mathcal{D}_{val}$, using the optimal incremental direction component $\mathcal{V}^*(\mathcal{M})$ that was learned at the lower level. This results in a validation loss $\mathcal{L}_{val}(\mathcal{V}^*(\mathcal{M}), \mathcal{M}) = C(\mathcal{V}^*(\mathcal{M}), \mathcal{M}; \mathcal{D}_{val})$. We determine the optimal magnitude component $\mathcal{M}$ by minimizing this validation loss:

$$\min_{\mathcal{M}} \mathcal{L}_{val}(\mathcal{V}^*(\mathcal{M}), \mathcal{M}) \tag{5}$$

**A Bi-level Optimization Framework** Integrating the two levels of optimization problems, we have the following bi-level optimization framework:

$$\min_{\mathcal{M}} \ \mathcal{L}_{val}(\mathcal{V}^*(\mathcal{M}), \mathcal{M})$$
$$s.t. \quad \mathcal{V}^*(\mathcal{M}) = \arg \min_{\mathcal{V}} \ \mathcal{L}_{tr}(\mathcal{V}, \mathcal{M}) \tag{6}$$

---

**Algorithm 1:** BiDoRA

---

**Input:** Training dataset $\mathcal{D}_{tr}$ and validation dataset $\mathcal{D}_{val}$

1 Initialize trainable magnitude components $\mathcal{M} = \{m_k\}_{k=1}^n$ and low-rank incremental direction components $\mathcal{V} = \{\Delta V_k\}_{k=1}^n = \{\{A_k\}_{k=1}^n, \{B_k\}_{k=1}^n\}$

2 // Search Phase

3 **while** *not converged* **do**

4     Update magnitude $\mathcal{M}$ by descending $\nabla_{\mathcal{M}}\mathcal{L}_{val}(\mathcal{V} - \xi\nabla_{\mathcal{V}}\mathcal{L}_{tr}(\mathcal{V}, \mathcal{M}), \mathcal{M})$

5     Update direction $\mathcal{V}$ by descending $\nabla_{\mathcal{V}}\mathcal{L}_{tr}(\mathcal{V}, \mathcal{M})$

6 Derive the optimal magnitude $\mathcal{M}^* = \{m_k^*\}_{k=1}^n$

7 // Retraining Phase

8 Train $\mathcal{V}$ until converge using $\mathcal{D}_{tr} \bigcup \mathcal{D}_{val}$ and derive the optimal direction $\mathcal{V}^*$

**Output:** $\mathcal{V}^*$ and $\mathcal{M}^*$

---

Note that these two levels of optimization problems are mutually dependent on each other. The solution of the optimization problem at the lower level, $\mathcal{V}^*(\mathcal{M})$, serves as a parameter for the upper-level problem, while the optimization variable $\mathcal{M}$ at the upper level acts as a parameter for the lower-level problem. By solving these two interconnected problems jointly, we can learn the optimal magnitude component $\mathcal{M}^*$ and incremental direction matrices $\mathcal{V}^*$ in an end-to-end manner.

Two reasons exist behind the choice of setting the magnitude component as the upper level instead of the converse one: 1) In literature, the upper level usually has much fewer parameters than the lower level. In our case, the design of setting the magnitude of complexity $\mathcal{O}(k)$ as the upper level and the direction of complexity $\mathcal{O}(dr + kr)$ as the lower level is consistent with the common practice. 2) BiDoRA resembles the DARTS method (Liu et al., 2018) in neural architecture search where the subnets are selected by a selection variable. Specifically, the magnitude vector resembles a selection variable on the direction matrix by softly selecting each direction (subnets) via scaling.

**Optimization Algorithm**  We use a gradient-based optimization algorithm (Choe et al., 2023) to solve the bi-level optimization problem presented in Eq. (6). A significant challenge in this process is that precisely computing the gradient of the upper-level loss $\mathcal{L}_{val}$ with respect to the magnitude component $\mathcal{M}$ can be computationally prohibitive due to the lack of an analytical solution for $\mathcal{V}^*(\mathcal{M})$ at the lower-level optimization problem. To address this issue, we use the following one-step-unrolled approximation of $\mathcal{V}^*(\mathcal{M})$ inspired by previous work (Liu et al., 2018):

$$\nabla_{\mathcal{M}}\mathcal{L}_{val}(\mathcal{V}^*(\mathcal{M}), \mathcal{M}) \approx \nabla_{\mathcal{M}}\mathcal{L}_{val}(\mathcal{V} - \xi\nabla_{\mathcal{V}}\mathcal{L}_{tr}(\mathcal{V}, \mathcal{M}), \mathcal{M})$$

where $\xi$ is the learning rate at the lower level, and the one-step-unrolled model $\bar{\mathcal{V}} = \mathcal{V} - \xi\nabla_{\mathcal{V}}\mathcal{L}_{tr}(\mathcal{V}, \mathcal{M})$ is used as a surrogate for the optimal solution $\mathcal{V}^*(\mathcal{M})$. We then compute the approximated gradient as follows:

$$\nabla_{\mathcal{M}}\mathcal{L}_{val}(\mathcal{V} - \xi\nabla_{\mathcal{V}}\mathcal{L}_{tr}(\mathcal{V}, \mathcal{M}), \mathcal{M})$$

$$= \nabla_{\mathcal{M}}\mathcal{L}_{val}(\bar{\mathcal{V}}, \mathcal{M}) - \xi\nabla_{\mathcal{M}, \mathcal{V}}^2\mathcal{L}_{tr}(\mathcal{V}, \mathcal{M})\nabla_{\bar{\mathcal{V}}}\mathcal{L}_{val}(\bar{\mathcal{V}}, \mathcal{M}) \tag{7}$$

$$\approx \nabla_{\mathcal{M}}\mathcal{L}_{val}(\bar{\mathcal{V}}, \mathcal{M}) - \xi\frac{\nabla_{\mathcal{M}}\mathcal{L}_{tr}(\mathcal{V}^+, \mathcal{M}) - \nabla_{\mathcal{M}}\mathcal{L}_{tr}(\mathcal{V}^-, \mathcal{M})}{2\epsilon} \tag{8}$$

where $\epsilon$ is a small scalar and $\mathcal{V}^\pm = \mathcal{V} \pm \epsilon\nabla_{\bar{\mathcal{V}}}\mathcal{L}_{val}(\bar{\mathcal{V}}, \mathcal{M})$. Since directly computing the matrix-vector multiplication term in Eq. (7) is computationally expensive, we use finite difference to approximate this product as in Eq. (8), following Liu et al. (2018). As detailed in Algorithm 1, the incremental direction component $\mathcal{V}$ and the magnitude component $\mathcal{M}$ are updated using gradient descent iteratively until convergence. After acquiring the optimal magnitudes $\mathcal{M}^*$ through the process above, the incremental direction component $\mathcal{V}$ is retrained on the union of training and validation splits to achieve the best performance on downstream tasks, resulting in the final learned $\mathcal{V}^*$.

Table 1: RoBERTa$_{base/large}$ (R$_{b/l}$) with different fine-tuning methods on the GLUE benchmark. A higher value is better for all datasets. The best results are shown in **bold**.

| Method | Param | MNLI | SST-2 | MRPC | CoLA | QNLI | QQP | RTE | STS-B | Avg |
|---|---|---|---|---|---|---|---|---|---|---|
| R$_b$(FT) | 125.0M | 90.3 | 94.8 | 89.3 | 61.6 | 86.7 | 92.8 | 76.9 | 91.2 | 85.5 |
| R$_b$(Adapter) | 0.9 M | 86.5 | 94.0 | 88.4 | 58.8 | 92.5 | 89.1 | 71.2 | 89.9 | 83.8 |
| R$_b$(LoRA) | 0.15 M | 86.8 | 94.3 | 88.0 | 60.3 | **93.0** | 89.6 | 72.9 | 90.1 | 84.4 |
| R$_b$(DoRA) | 0.17 M | 86.8 | 94.2 | 89.2 | 60.5 | 92.9 | 89.6 | 73.2 | **90.2** | 84.6 |
| R$_b$(BiDoRA) | 0.17 M | **87.1** | **94.4** | **89.4** | 61.3 | 92.7 | 90.6 | **76.0** | 90.1 | **85.2** |
| R$_l$(FT) | 355.0M | 90.2 | 96.4 | 90.9 | 68.0 | 94.7 | 92.2 | 86.6 | 92.4 | 88.9 |
| R$_l$(Adapter) | 0.8M | 90.3 | 96.3 | 87.7 | 66.3 | **94.7** | 91.5 | 72.9 | 91.5 | 86.4 |
| R$_l$(LoRA) | 0.39 M | **90.6** | 96.3 | 90.0 | 66.9 | 94.5 | 91.2 | 86.3 | 91.7 | 88.4 |
| R$_l$(DoRA) | 0.39 M | **90.6** | **96.4** | 89.8 | 65.8 | **94.7** | 91.2 | 86.6 | **92.0** | 88.4 |
| R$_l$(BiDoRA) | 0.39 M | **90.6** | 96.1 | **90.1** | 67.0 | 94.6 | **91.7** | **86.9** | **92.0** | **88.6** |

## 5 EXPERIMENTS

### 5.1 EXPERIMENTAL SETUP

We compare BiDoRA with several PEFT methods, including Adapter tuning (Houlsby et al., 2019), LoRA (Hu et al., 2021), and DoRA (Liu et al., 2024). BiDoRA does not use any additional data compared to other baselines, as we create the validation set for upper-level optimization by splitting the original training set with an 8:2 ratio for all tasks. All methods in the experiment, including ablation studies, are trained until convergence for a fair comparison. Detailed descriptions of these baseline methods are provided in Appendix C.

Our experiments cover a wide range of tasks, including natural language understanding (NLU), natural language generation (NLG), and token classification. For NLU tasks, we fine-tune the RoBERTa-base and RoBERTa-large models on the GLUE benchmark (Wang et al., 2019) and the Reuters21578 dataset (Padmanabhan et al., 2016) using all baseline PEFT methods and BiDoRA. Detailed descriptions of these datasets and pre-trained models are provided in Appendix A. Following existing practices, the development set is used in GLUE as the test data since the actual test set is not publicly available. We report the overall (matched and mismatched) accuracy for MNLI, Matthew's correlation for CoLA, Pear-

Table 2: RoBERTa$_{base/large}$ (R$_{b/l}$) with different fine-tuning methods on the Reuters21578 benchmark. A higher value is better for all datasets. The best results are shown in **bold**.

| Method | Param | ModApte | ModHayes | ModLewis |
|---|---|---|---|---|
| R$_b$(FT) | 125.0M | 85.4 | 77.6 | 77.1 |
| R$_b$(Adapter) | 0.9 M | **85.3** | 77.5 | 76.8 |
| R$_b$(LoRA) | 0.15 M | 84.7 | 74.3 | 74.7 |
| R$_b$(DoRA) | 0.17 M | 84.8 | 78.2 | 76.6 |
| R$_b$(BiDoRA) | 0.17 M | **85.3** | **79.9** | **77.6** |
| R$_l$(FT) | 355.0M | 84.8 | 77.5 | 76.6 |
| R$_l$(Adapter) | 0.44 M | 84.8 | 77.9 | 76.7 |
| R$_l$(LoRA) | 0.39 M | 84.7 | 77.7 | 76.7 |
| R$_l$(DoRA) | 0.39 M | 84.8 | 77.4 | 76.7 |
| R$_l$(BiDoRA) | 0.39 M | **84.9** | **78.9** | **77.3** |

son correlation for STS-B, and accuracy for the other tasks. On the Reuters21578 dataset, the F1 score is used as the evaluation metric across all three splits. For NLG tasks, we fine-tune GPT-2 medium on the E2E (Novikova et al., 2017) dataset. We use BLEU (Papineni et al., 2002), NIST (Lin & Och, 2004), METEOR (Banerjee & Lavie, 2005), ROUGE-L (Lin, 2004), and CIDEr (Vedantam et al., 2015) as evaluation metrics. For token classification, we fine-tune the RoBERTa-base and RoBERTa-large models on the BioNLP (Collier et al., 2004) dataset and the CoNLL2003 (Sang & De Meulder, 2003) dataset. Accuracy, precision, recall, and F1 score are used as evaluation metrics.

For all experiments, our implementation is based on the Huggingface Transformers library (Wolf et al., 2019) and the Betty library (Choe et al., 2023). We use a single NVIDIA A100 GPU for all experiments. More detailed experimental settings are provided in Appendix B.

### 5.2 EXPERIMENTS ON NATURAL LANGUAGE UNDERSTANDING TASKS

In this section, we evaluate the performance of BiDoRA on NLU tasks, with a particular focus on text classification. Table 1 presents the results of fine-tuning the RoBERTa-base and RoBERTa-

Table 4: RoBERTa$_{base/large}$ (R$_{b/l}$) with different fine-tuning methods on BioNLP data and CoNLL2003 dataset. A higher value is better for all metrics. The best results are shown in **bold**.

| Method | Param | BioNLP | | | | CoNLL2003 | | | |
|---|---|---|---|---|---|---|---|---|---|
| | | Accuracy | Precision | Recall | F1 | Accuracy | Precision | Recall | F1 |
| R$_b$(FT) | 125.0M | 93.9 | 69.0 | 78.9 | 73.6 | 99.3 | 95.7 | 96.3 | 96.0 |
| R$_b$(Adapter) | 0.9 M | 93.9 | 69.1 | 78.8 | 73.7 | **99.3** | 95.7 | 96.4 | 96.0 |
| R$_b$(LoRA) | 0.15 M | 93.9 | 69.0 | 78.8 | 73.6 | **99.3** | 95.4 | 96.3 | 95.8 |
| R$_b$(DoRA) | 0.17 M | **94.0** | 69.2 | **79.1** | 73.8 | **99.3** | 95.3 | 96.2 | 95.8 |
| R$_b$(BiDoRA) | 0.17 M | 93.9 | **71.2** | 78.6 | **74.7** | **99.3** | **95.9** | **96.5** | **96.2** |
| R$_l$(FT) | 355.0M | 94.0 | 69.4 | 79.6 | 74.1 | 99.4 | 96.2 | 97.0 | 96.6 |
| R$_l$(Adapter) | 0.44 M | **94.0** | 69.4 | 79.7 | 74.2 | **99.4** | 96.1 | 97.0 | 96.6 |
| R$_l$(LoRA) | 0.39 M | 93.9 | 69.2 | 79.3 | 73.9 | **99.4** | 96.2 | 97.0 | 96.6 |
| R$_l$(DoRA) | 0.39 M | **94.0** | 69.4 | **79.7** | 74.2 | **99.4** | 96.2 | **97.1** | 96.6 |
| R$_l$(BiDoRA) | 0.39 M | **94.0** | **71.3** | 79.3 | **75.1** | **99.4** | **96.4** | **97.1** | **96.7** |

large models on the GLUE benchmark with baseline PEFT methods and BiDoRA. The results show that BiDoRA achieves superior or comparable performance compared to baseline methods across all datasets with the same number of trainable parameters. Table 2 presents the results of fine-tuning RoBERTa models on the Reuters21578 datasets, where BiDoRA outperforms all baseline methods by an even larger margin. Notably, BiDoRA achieves performance comparable to or even better than full fine-tuning. The superior performance of BiDoRA on both benchmarks verifies the effectiveness of its bi-level optimization mechanism. By training the magnitude and incremental direction components on two distinct sub-datasets, BiDoRA enhances the flexibility of the learning process and improves learning capacity compared to DoRA, resulting in a performance boost.

## 5.3 EXPERIMENTS ON NATURAL LANGUAGE GENERATION TASKS

In this section, we evaluate BiDoRA's performance on the NLG task. Table 3 presents the results of fine-tuning a GPT-2 model on the E2E dataset with baseline PEFT methods and BiDoRA. The results show that BiDoRA achieves the best performance across all five evaluation metrics, demonstrating the superiority of BiDoRA in fine-tuning pre-trained models for NLG tasks.

## 5.4 EXPERIMENTS ON TOKEN CLASSIFICATION

Further evidence of the effectiveness of BiDoRA can be observed in Table 4, which reports the results of token classification tasks. Unlike the NLU tasks discussed in the previous section, which involve classifying entire sentences and

Table 3: Performance of BiDoRA and baseline methods for fine-tuning GPT2-medium on the E2E dataset. A higher value is better for all metrics. The best results are shown in **bold**.

| Method | Param | BLEU | NIST | MET | ROUGE-L | CIDEr |
|---|---|---|---|---|---|---|
| Full FT | 354.9M | 68.0 | 8.61 | 46.1 | 69.0 | 2.38 |
| Adapter | 11.1M | 67.0 | 8.50 | 45.2 | 66.9 | 2.31 |
| LoRA | 0.39M | 67.1 | 8.54 | 45.7 | 68.0 | 2.33 |
| DoRA | 0.39M | 67.0 | 8.48 | 45.4 | 70.1 | 2.33 |
| BiDoRA | 0.39M | **69.0** | **8.72** | **46.2** | **70.9** | **2.44** |

Table 5: Fine-tuning ESM on the thermostability prediction task. A higher value is better for all metrics, with the best results highlighted in **bold**.

| Methods | #Params | Accuracy | Precision | Recall | F1 |
|---|---|---|---|---|---|
| FT | 652.7M | 79.8 | 81.2 | 79.8 | 78.4 |
| LoRA | 1.5M | 75.9 | 78.2 | 75.9 | 75.5 |
| DoRA | 1.6M | 76.9 | 78.7 | 76.9 | 76.2 |
| BiDoRA | 1.6M | **78.8** | **79.1** | **78.8** | **78.2** |

focusing on capturing global semantics, token classification requires classifying each token within a sentence, highlighting the importance of capturing local context. On the BioNLP dataset, BiDoRA consistently outperforms baseline methods by a large margin in terms of F1 score. On the CoNLL2003 dataset, BiDoRA either outperforms or matches all baseline methods across all metrics. Consistent with our previous findings, BiDoRA effectively fine-tunes pre-trained models for token classification tasks.

## 5.5 EXPERIMENTS ON EXTREMELY SMALL DATASETS

The ESM (Evolutionary Scale Modeling, Rives et al. (2021)) model is a transformer-based protein language model designed for protein sequence analysis, leveraging the transformer architecture to capture evolutionary patterns. We fine-tune the ESM model using the Protein Aligner checkpoint (Zhang et al., 2024a) on three protein prediction tasks: two classification tasks—thermostability prediction (Chen et al., 3,695 training samples) and blood-brain barrier peptide prediction (BBP, Dai et al. (2021), 936 training samples)—and one regression task, minimum inhibitory concentration prediction (MIC, Ledesma-Fernandez et al. (2023), 200 training samples). Notably, protein analysis datasets are typically much smaller than those in NLP, in which case the large pre-trained models are prone to overfitting, even when using PEFT methods. The trainable parameters (on the order of millions) are significantly overparameterized compared to the available samples (thousands or even hundreds), highlighting the need for our overfitting-resilient counterpart. The results are presented in Tables 5, 6, and 7, respectively. For the classification tasks, we use accuracy, precision, recall, and F1 score to evaluate performance. For the regression task, we use mean squared error (MSE). Consistent with our previous findings, BiDoRA effectively fine-tunes pre-trained models on extremely small datasets. Our method outperforms the baselines by a larger margin as the dataset size decreases, confirming our previous conclusion that our method effectively combats the overfitting issue on various network architectures and diverse tasks.

Table 6: Fine-tuning ESM on the BBP task. A higher value is better for all metrics, with the best results highlighted in **bold**.

| Methods | #Params | Accuracy | Precision | Recall | F1 |
|---|---|---|---|---|---|
| FT | 652.9M | 89.4 | 89.9 | 89.4 | 89.4 |
| LoRA | 1.9M | 86.8 | 87.7 | 86.8 | 86.7 |
| DoRA | 2.0M | 89.4 | 91.3 | 89.4 | 89.3 |
| BiDoRA | 2.0M | **92.1** | **93.1** | **92.1** | **92.0** |

## 5.6 Scaling Up to Larger Model Settings

We use DeBERTa (He et al., 2020) and Llama2 (Touvron et al., 2023) to evaluate the scalability of our method. For our experiments, we use DeBERTa-v2-xxlarge, which has 1.5 billion parameters, and Llama2-7b, which has 7 billion parameters. We evaluate the three subsets of the Reuters21578 dataset. The results presented in Table 8 show that BiDoRA achieves better or on-par performance compared with DoRA and full fine-tuning (FT), indicating that BiDoRA yields better generalization when fine-tuning models with a very large number of parameters and diverse network architectures.

Table 7: Fine-tuning ESM on the MIC task. A lower value is better, with the best results highlighted in **bold**.

| Methods | #Params | MSE |
|---|---|---|
| FT | 652.7M | 0.2894 |
| LoRA | 1.7M | 0.3433 |
| DoRA | 1.8M | 0.2918 |
| BiDoRA | 1.8M | **0.2818** |

## 5.7 Ablation Studies

In this section, we perform ablation studies to investigate the effectiveness of individual modules or strategies in BiDoRA. We fine-tune a RoBERTa-base model on the GLUE benchmark under different ablation settings, and the results are shown in Table 9.

Table 8: Fine-tuning DeBERTa and Llama2 on the Reuters21578 benchmark. A higher value is better for all datasets, with the best results highlighted in **bold**.

| Method | Param | ModApte | ModHayes | ModLewis |
|---|---|---|---|---|
| DeBERTa(DoRA) | 1.3M | 79.3 | 75.4 | 73.6 |
| DeBERTa(BiDoRA) | 1.3M | **79.9** | **75.7** | **74.5** |
| Llama2-7b (DoRA) | 2.4M | 81.8 | 76.4 | 74.5 |
| Llama2-7b (BiDoRA) | 2.4M | **82.4** | **77.1** | **74.8** |

**Retraining**   We test the model directly obtained from the search phase to evaluate the effectiveness of further retraining the incremental direction component. The results show that BiDoRA outperforms BiDoRA (w/o retraining) on average, highlighting the necessity of retraining.

**Bi-level Optimization**   We set $\xi$ to zero in Algorithm 1 to assess the effectiveness of the bi-level optimization framework. This ablation setting can be interpreted as an alternative learning method where two optimization steps are carried out alternately on two different splits of the training dataset. Notably, in the alternative learning method, the updating of each component is unaware of each other, making the training less stable. In contrast, the hyper-gradient used in bi-level optimization avoids this issue by connecting the two levels in a certain way. The results show that BiDoRA outperforms BiDoRA ($\xi = 0$) on average, demonstrating the efficacy of the bi-level optimization strategy.

Table 9: Ablation studies. We evaluate the performance of BiDoRA without retraining (w/o retraining), without bi-level optimization ($\xi = 0$) and without orthogonal regularization (w/o cst.).

| Method | MNLI | SST-2 | MRPC | CoLA | QNLI | QQP | RTE | STS-B | Avg |
|---|---|---|---|---|---|---|---|---|---|
| BiDoRA (w/o retraining) | 87.0 | 94.2 | 89.0 | 57.3 | 92.4 | 90.6 | 71.6 | 90.0 | 84.0 |
| BiDoRA ($\xi = 0$) | 86.9 | 94.2 | 89.0 | 59.4 | 90.8 | **91.2** | 75.9 | 90.0 | 84.7 |
| BiDoRA (w/o cst.) | 87.0 | **94.4** | 88.6 | **61.3** | **92.7** | 90.2 | 76.0 | **90.1** | 85.0 |
| BiDoRA | **87.1** | **94.4** | **89.4** | **61.3** | **92.7** | 90.6 | **76.1** | **90.1** | **85.2** |

**Orthogonal Regularization** We examine the effectiveness of the orthogonality constraint in Eq. (3) by setting $\gamma$ to zero. Results show that BiDoRA outperforms BiDoRA (w/o cst.) on average, indicating the effectiveness of applying the orthogonality regularizer to alleviate overfitting.

### 5.8 WEIGHT DECOMPOSITION ANALYSIS

One important motivation of DoRA is to bridge the inherent differences between LoRA and FT. Similar to DoRA, we conduct a weight decomposition analysis on the correlation between the change of magnitudes and that of directions (detailed in Appendix E) for BiDoRA and baseline methods by fine-tuning a GPT2-medium model on the E2E dataset. As shown in Figure 3, FT, DoRA, and BiDoRA all exhibit negative correlation values, while LoRA shows a positive correlation, consistent with the findings in Liu et al. (2024). Notably, BiDoRA achieves a negative correlation of $-8.042$, closer to FT than DoRA's $-1.784$. This improvement is attributed to the decoupled training process of the two layers, which allows for a higher learning capacity compared to DoRA.

### 5.9 COMPUTATION COSTS

Since BiDoRA has the same architecture as DoRA, our method only requires two extra forward and backward passes of the lower level for the hypergradient calculation of the upper level, as shown in Eq. 8. In principle, this would make our method roughly three-fold computationally costly. A similar analysis holds for the memory consumption analysis. Empirically, table 10 shows the average training cost of BiDoRA and two baseline methods on the MNLI, QQP, and SST-2 datasets from the GLUE benchmark, being consistent with the theoretical analysis. We normalize the cost of LoRA to 1 for reference. Importantly, we also found that BiDoRA converges significantly faster than DoRA, as indicated by the orange dashed lines in Figure 4, mitigating the speed disadvantage. This allows BiDoRA to achieve comparable performance in the same wall-clock time. Therefore, BiDoRA is practical due to its superior performance and comparable speed.

Table 10: Average training time cost on the MNLI, QQP, and SST-2 datasets.

| Method | LoRA | DoRA | BiDoRA |
|---|---|---|---|
| Cost | $\times 1$ | $\times 1.30$ | $\times 3.92$ |

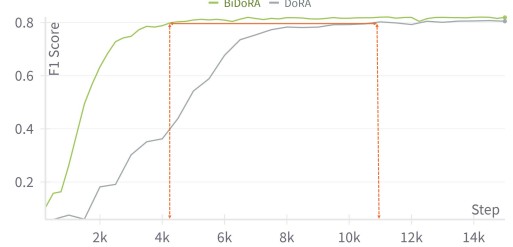

Figure 4: BiDoRA reaches a comparable performance using much fewer iterations than DoRA. Evaluated when fine-tuning the Llama2-7b model on the Reuters-21578 dataset, and a similar pattern holds for other datasets.

## 6 CONCLUSION AND FUTURE WORKS

We propose BiDoRA, a novel bi-level optimization framework for parameter-efficient fine-tuning of large-scale pre-trained models. By conducting weight decomposition following the DoRA approach, our method trains the two components separately in two interconnected optimization levels using different sub-datasets. In this way, BiDoRA not only decouples the learning process of the two components, resulting in a learning pattern closer to FT, but also effectively alleviates overfitting. Empirical studies on various NLP tasks demonstrate that BiDoRA outperforms DoRA and other baselines, highlighting the effectiveness of our method. We leave enhancing the efficiency of BiDoRA for future work. Overall computational costs can be reduced by using more accurate hyper-gradient estimators, such as implicit differentiation (Rajeswaran et al., 2019) and the SAMA estimator (Choe et al., 2024), which can be easily integrated into our framework thanks to support from the Betty library (Choe et al., 2023).

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

# A  DATASETS AND MODELS

## A.1  NATURAL LANGUAGE UNDERSTANDING

The GLUE Benchmark comprises a diverse array of tasks that are widely employed for evaluation in natural language understanding. It encompasses two single-sentence classification tasks, three tasks assessing similarity and paraphrasing, and four tasks focusing on natural language inference. Specifically, it includes MNLI (MultiNLI, Williams et al. (2017)), SST-2 (Stanford Sentiment Treebank, Socher et al. (2013)), MRPC (Microsoft Research Paraphrase Corpus, Dolan & Brockett (2005)), CoLA (Corpus of Linguistic Acceptability, Warstadt et al. (2019)), QNLI (Question NLI, Rajpurkar et al. (2018)), QQP (Quora Question Pairs), RTE (Recognizing Textual Entailment), and STS-B (Semantic Textual Similarity Benchmark, Cer et al. (2017)). We summarize the statistical data for all datasets within the GLUE Benchmark in Table 11.

The Reuters-21578 (Padmanabhan et al., 2016) dataset is one of the most widely used data collections for text categorization research. It was collected from the Reuters financial newswire service in 1987 and is used for text classification and natural language processing tasks. Three splits are available: ModApte, ModHayes, and ModLewis. These documents cover various topics, such as politics, economics, and sports. We summarize the statistical data for all text classification tasks used in our experiments in Table 12.

Table 11: The statistical data for all datasets within the GLUE Benchmark.

| Dataset | Metrics | Train | Dev | Test | Label | Task |
|---|---|---|---|---|---|---|
| MNLI | Accuracy | 393k | 20k | 20k | 3 | NLI |
| SST-2 | Accuracy | 67k | 872 | 1.8k | 2 | Sentiment |
| MRPC | Accuracy | 3.7k | 408 | 1.7k | 2 | Paraphrase |
| CoLA | Matthews Corr | 8.5k | 1k | 1k | 2 | Acceptability |
| QNLI | Accuracy | 108k | 5.7k | 5.7k | 2 | QA/NLI |
| QQP | Accuracy | 364k | 40k | 391k | 2 | Paraphrase |
| RTE | Accuracy | 2.5k | 276 | 3k | 2 | NLI |
| STS-B | Pearson Corr | 7.0k | 1.5k | 1.4k | 1 | Similarity |

Table 12: The statistical data for the Reuters-21578 dataset.

| Dataset | Metrics | Train | Test |
|---|---|---|---|
| ModApte | F1 | 8.8k | 3k |
| ModHayes | F1 | 18k | 0.7k |
| ModLewis | F1 | 12k | 5.5k |

## A.2  NATURAL LANGUAGE GENERATION

In our experiments on natural language generation, we use the E2E (Novikova et al., 2017) dataset, which was initially introduced as a dataset for training end-to-end, data-driven natural language generation systems. Multiple references can be associated with each source table used as input. Each sample input $(x, y)$ consists of a series of slot-value pairs accompanied by an associated natural language reference text. The E2E dataset comprises approximately 42,000 training examples, 4,600 validation examples, and 4,600 test examples from the restaurant domain.

We utilize the following five evaluation metrics: BLEU (Papineni et al., 2002), NIST (Lin & Och, 2004), METEOR (Banerjee & Lavie, 2005), ROUGE-L (Lin, 2004), and CIDEr (Vedantam et al., 2015). We summarize its statistical data in Table 13.

Table 13: The statistical data for E2E dataset.

| Dataset | Metrics | Train | Validation |
|---------|---------|-------|------------|
| E2E | BLEU,NIST,MET,ROUGE-L,CIDEr | 42k | 4.6k |

## A.3 TOKEN CLASSIFICATION

BioNLP (Collier et al., 2004) is a Named Entity Recognition dataset that contains biological entities such as DNA, RNA, and protein. It is essentially a token classification task where we want to classify each entity in the sequence. CoNLL-2003 (Sang & De Meulder, 2003) focuses on language-independent named entity recognition. It concentrates on four types of named entities: persons, locations, organizations, and miscellaneous entities that do not belong to the previous three groups. We summarize the statistical data for all used token classification tasks in Table 14.

Table 14: The statistical data for token classification tasks

| Dataset | Metrics | Train | Validation | Test |
|---------|---------|-------|------------|------|
| BioNLP | Accuracy,Precision,Recall,F1 | 17k | 1.9k | 3.9k |
| CoNLL2003 | Accuracy,Precision,Recall,F1 | 14k | 3.3k | 3.5k |

## B EXPERIMENTAL SETTINGS

In this section, we provide detailed experimental settings. We maintain consistent configurations across experiments, including LoRA rank, LoRA $\alpha$, batch size, maximum sequence length, and optimizer, to ensure a fair comparison. The hyperparameter tuning for our method is straightforward and convenient.

### B.1 ROBERTA

We summarize the experimental settings for the GLUE benchmark in Table 15 and for the Reuters21578 dataset and token classification tasks in Table 16.

### B.2 GPT-2

We summarize the experimental settings for the GPT-2 experiments in Table 17. The experimental configuration, particularly during the inference stage, follows the approach described by Hu et al. (2021).

## C BASELINES IN EXPERIMENTS

We compare BiDoRA with Full Fine-Tuning (FT), Adapter tuning (Houlsby et al., 2019), LoRA (Hu et al., 2021), and DoRA (Liu et al., 2024) in all our experiments. We provide a brief introduction to these methods here.

**Full Fine-Tuning (FT)** is a commonly used method for adaptation. The model is initialized with pre-trained weights and biases, and all model parameters are updated through gradient descent.

**Adapter tuning** (Houlsby et al., 2019) inserts layer adapters between neural modules, such as the MLP module or the self-attention module. It incorporates two fully connected layers within an adapter layer, with a nonlinearity function applied between them.

Table 15: The hyperparameters we used for RoBERTa on the GLUE benchmark.

| Method | Settings | MNLI | SST-2 | MRPC | CoLA | QNLI | QQP | RTE | STS-B |
|---|---|---|---|---|---|---|---|---|---|
| | Optimizer | | | | AdamW | | | | |
| | Warmup Ratio | | | | 0.06 | | | | |
| | Scheduler | | | | Linear | | | | |
| | LoRA rank | | | | $\text{rank}_a = \text{rank}_u = 4$ | | | | |
| | LoRA $\alpha$ | | | | 8 | | | | |
| RoBERTa-base | Total batch size | | | | 32 | | | | |
| | Global steps | 20000 | 12000 | 25000 | 20000 | 15000 | 20000 | 15000 | 12000 |
| | Lower learning rate | 5e-5 | 1e-5 | 2e-5 | 5e-5 | 2e-5 | 5e-5 | 1e-5 | 1e-5 |
| | Upper learning rate | 5e-5 | 1e-5 | 2e-5 | 5e-5 | 2e-5 | 5e-5 | 1e-5 | 1e-5 |
| | Lower weight decay | | | | 0.1 | | | | |
| | Upper weight decay | 0.1 | 0.1 | 0.1 | 0.1 | 0 | 0.1 | 0.1 | 0.01 |
| | Max Seq Length | | | | 512 | | | | |
| | Regularization Coefficient | | | | 1e-5 | | | | |
| RoBERTa-large | Total batch size | | | | 32 | | | | |
| | Global steps | 50000 | 20000 | 30000 | 20000 | 60000 | 40000 | 15000 | 10000 |
| | Lower learning rate | | | | 1e-5 | | | | |
| | Upper learning rate | | | | 1e-5 | | | | |
| | Lower weight decay | 0.5 | 0.5 | 0 | 0.2 | 0.5 | 0.5 | 0.5 | 0.5 |
| | Upper weight decay | 0.5 | 0.05 | 0 | 0.2 | 0.5 | 0.5 | 0.1 | 0.5 |
| | Max Seq Length | | | | 128 | | | | |
| | Regularization Coefficient | 0 | 0 | 1e-5 | 1e-5 | 0 | 1e-5 | 0 | 1e-5 |

Table 16: The hyperparameters we used for RoBERTa on the Reuters21578 dataset, BioNLP dataset, and CoNLL2003 dataset.

| Method | Settings | ModApte | ModHayes | ModLewis | BioNLP | CoNLL2003 |
|---|---|---|---|---|---|---|
| | Optimizer | | | AdamW | | |
| | Warmup Ratio | | | 0.06 | | |
| | Scheduler | | | Linear | | |
| | LoRA rank | | | $\text{rank}_a = \text{rank}_u = 4$ | | |
| | LoRA $\alpha$ | | | 8 | | |
| RoBERTa-base | Total batch size | | | 32 | | |
| | Global steps | 20000 | 20000 | 20000 | 12000 | 12000 |
| | Lower learning rate | 3e-5 | 3e-5 | 3e-5 | 1e-5 | 2e-5 |
| | Upper learning rate | 3e-5 | 3e-5 | 3e-5 | 1e-5 | 2e-5 |
| | Lower weight decay | 0.1 | 0.1 | 0.1 | 0.1 | 0.2 |
| | Upper weight decay | | | 0.1 | | |
| | Max Seq Length | | | 512 | | |
| | Regularization Coefficient | 0 | 1e-5 | 0 | 1e-5 | 0 |
| RoBERTa-large | Total batch size | | | 32 | | |
| | Global steps | 20000 | 20000 | 20000 | 12000 | 15000 |
| | Lower learning rate | 1e-5 | 1e-5 | 1e-5 | 2e-5 | 1e-5 |
| | Upper learning rate | 1e-5 | 1e-5 | 1e-5 | 2e-5 | 1e-5 |
| | Lower weight decay | 0.2 | 0.1 | 0.2 | 0.02 | 0.1 |
| | Upper weight decay | 0.1 | 0.1 | 0.1 | 0.02 | 0.1 |
| | Max Seq Length | | | 128 | | |
| | Regularization Coefficient | 0 | 1e-5 | 0 | 0 | 1e-5 |

**LoRA** (Hu et al., 2021) adds trainable incremental update matrices to pre-trained weight matrices. Following the experimental settings of LoRA, we applied BiDoRA to $W_q$ and $W_v$ matrices (the query and value weight matrices in the self-attention module) for a fair comparison.

Table 17: The hyperparameters we used for GPT-2 on the E2E NLG benchmark.

| Settings | Training |
|---|---|
| Optimizer | AdamW |
| Warmup Ratio | 0.06 |
| Scheduler | Linear |
| LoRA rank | $\text{rank}_a = \text{rank}_u = 4$ |
| LoRA $\alpha$ | 32 |
| Label Smooth | 0.1 |
| Lower learning rate | 1e-3 |
| Upper learning rate | 1e-4 |
| Lower weight decay | 1 |
| Upper weight decay | 1 |
| Max Seq Length | 512 |
| Regularization Coefficient | 1e-5 |

| Settings | Inference |
|---|---|
| Beam Size | 10 |
| Length Penalty | 0.9 |
| no repeat ngram size | 4 |

**DoRA** (Liu et al., 2024) proposes weight-decomposed adaptation, which formulates the incremental matrices as a product of magnitude and direction components, thereby accelerating training and aligning the training behavior with full fine-tuning. In contrast, our BiDoRA trains the two components on distinct sub-datasets to alleviate overfitting.

## D  THEORETICAL INSIGHTS

We can gain several insights into how BiDoRA can alleviate the overfitting issue by analyzing both the gradient of the bi-level optimization and theoretical results from existing work.

In terms of gradient analysis, we can demonstrate the robustness of our method by examining the behavior of the loss $\mathcal{L}'_{val}(\mathcal{V}, \mathcal{M}) = \mathcal{L}_{val}(\mathcal{V} - \xi \nabla_{\mathcal{V}} \mathcal{L}_{tr}(\mathcal{V}, \mathcal{M}), \mathcal{M})$. We analyze its gradient with respect to the magnitude parameters $\mathcal{M}$ using the chain rule,

$$\frac{d\mathcal{L}'_{val}}{d\mathcal{M}} = \frac{\partial \mathcal{L}'_{val}}{\partial \mathcal{M}} + \frac{\partial \bar{\mathcal{V}}}{\partial \mathcal{M}} \times \frac{\partial \mathcal{L}'_{val}}{\partial \bar{\mathcal{V}}} \tag{9}$$

The second term on the right-hand side, particularly the term $\frac{\partial \bar{\mathcal{V}}}{\partial \mathcal{M}}$, referred to as the best-response Jacobian in the literature, captures how the direction component would react to changes in the magnitude. More specifically, the update of $\mathcal{M}$ must consider not only the current value of the direction $\mathcal{V}$ but also additional information about how the direction responds to changes in the magnitude. This facilitates finding a globally optimal magnitude value, thereby enhancing stability and robustness. Consequently, the model becomes more resilient to overfitting. In contrast, if $\xi$ is set to 0, Algorithm 1 reduces to a first-order approximation, which performs coordinate descent for $\mathcal{M}$ and $\mathcal{V}$ on different empirical distributions. In this scenario, the second term in Eq. (9) vanishes, making the approach insufficient to alleviate overfitting effectively.

On the other hand, we can also see the generalization guarantees of BiDoRA using various theoretical works in the literature (Oymak et al., 2021; Bai et al., 2021; Bao et al., 2021; Chen et al., 2022; Huang et al., 2022). For example, Theorem 1 in Oymak et al. (2021) provides strong evidence of the benefits of applying bi-level optimization to alleviate overfitting. Oymak et al. (2021) discussed the generalization guarantees for neural architecture search with a train-validation split. They assumed the same scenario as ours: using bi-level optimization where one optimizes the weights over the training data (lower-level problem) and hyperparameters over the validation data (upper-level problem). The theorem shows that as soon as the size of the validation data exceeds the effective number of hyperparameters (up to logarithmic factors), the test error is close to the validation error (i.e., validation error is indicative of the test error). In practice, the number of magnitude parameters is generally less than or comparable to the size of the validation set, making this assumption

feasible. We empirically show that validation performance and test performance align, as theoretically supported by Oymak et al. (2021). In Figure 5, we plot the performance on the training, validation, and test datasets for the CoLA and SST-2 datasets during the search phase. The first observation is that training performance always overfits until reaching zero error, whereas validation performance closely tracks test performance. This training behavior aligns with the fact that large-capacity networks can perfectly fit (Du et al., 2019; Ji & Telgarsky, 2019; Oymak & Soltanolkotabi, 2020) in the overparameterized setting, where the available data (less than 10k for CoLA and 67k for SST-2) is less than the number of learnable parameters (around 170k). To truly find a model that achieves good generalization performance, the optimization procedure should evaluate the generalization loss. Thus, as shown in Figure 5, the validation phase serves as a crucial test proxy in the overparameterized setting where training performance may not be indicative of test performance.

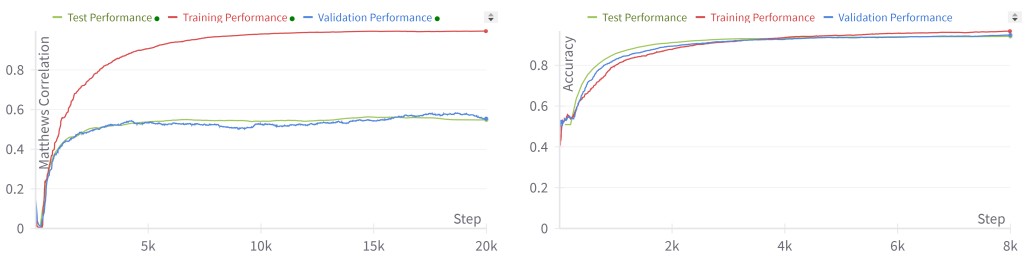

Figure 5: Train/Validation/Test performance during the search phase of fine-tuning the RoBERTa-base model on the CoLA dataset (left) and SST-2 (right).

## E  WEIGHT DECOMPOSITION ANALYSIS

We provide a brief review of the weight decomposition analysis proposed in Liu et al. (2024). Define the weight decomposition of a weight matrix $W \in \mathbb{R}^{d \times k}$ (e.g., query matrix in an attention layer) as $W = m \frac{V}{\|V\|_c} = \|W\|_c \frac{W}{\|W\|_c}$, where $m \in \mathbb{R}^{1 \times k}$ is the magnitude vector, and $V \in \mathbb{R}^{d \times k}$ is the directional matrix, with $\| \cdot \|_c$ representing the vector-wise norm of a matrix across each column. This decomposition ensures that each column of $V/\|V\|_c$ remains a unit vector, and the corresponding scalar in $m$ defines the magnitude of each vector. Liu et al. (2024) examine the magnitude and directional variations between $W_0$ and $W_{\mathrm{FT}}$, defined as $\Delta M_{\mathrm{FT}}^t = \frac{\sum_{n=1}^k |m_{\mathrm{FT}}^{n,t} - m_0^n|}{k}$ and $\Delta D_{\mathrm{FT}}^t = \frac{\sum_{n=1}^k (1 - \cos(V_{\mathrm{FT}}^{n,t}, W_0^n))}{k}$. Here, $\Delta M_{\mathrm{FT}}^t$ and $\Delta D_{\mathrm{FT}}^t$ represent the magnitude and direction differences between $W_0$ and $W_{\mathrm{FT}}$ at the $t$-th training step, respectively, with $\cos(\cdot, \cdot)$ denoting cosine similarity. $m_{\mathrm{FT}}^{n,t}$ and $m_0^n$ are the $n^{th}$ scalars in their respective magnitude vectors, while $V_{\mathrm{FT}}^{n,t}$ and $W_0^n$ are the $n^{th}$ columns in $V_{\mathrm{FT}}^t$ and $W_0$. Intuitively, a consistent positive slope trend across all the intermediate steps implies a difficulty in concurrent learning of both magnitude and direction, suggesting that slight directional changes are challenging to execute alongside more significant magnitude alterations. In contrast, a relatively negative slope signifies a more varied learning pattern, with a more pronounced negative correlation indicating a larger learning capacity.

Complementary to Figure 3 in the main paper on the query matrix, we provide additional results of weight decomposition analysis in Figure 6 on the value matrix to complement the findings in Section 5.8. We can draw two key observations from Figure 6: 1) Consistent with the results in Liu et al. (2024), both FT and DoRA exhibit negative correlation values of $-49.279$ and $-5.485$, respectively, while LoRA shows a positive correlation with a value of $2.503$. 2) BiDoRA achieves a negative correlation value of $-10.547$, indicating closer alignment with FT compared to DoRA. The analysis of how BiDoRA achieves this improvement is similar to that discussed in Section 5.8.

## F  THE ROLE OF HYPERPARAMETER

The hyperparameter tuning for BiDoRA is simple, convenient, and straightforward. We further conducted experiments regarding the dataset partition of $\mathcal{D}_{tr}$ and $\mathcal{D}_{val}$ to provide insights into its role in BiDoRA. The dataset partition helps maintain the balance of inner/outer optimization by

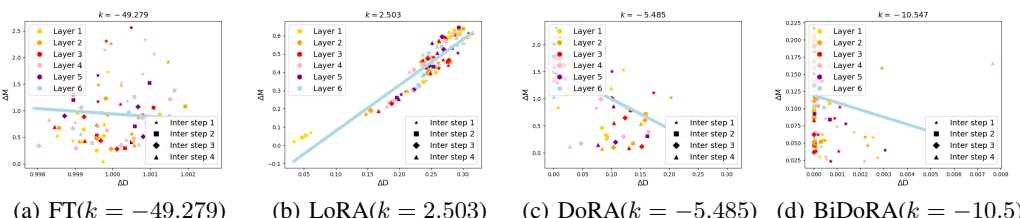

(a) FT($k = -49.279$)    (b) LoRA($k = 2.503$)    (c) DoRA($k = -5.485$)    (d) BiDoRA($k = -10.5$)

Figure 6: **Magnitude and direction updates** for (a) FT, (b) LoRA, (c) DoRA, and (d) BiDoRA of the value matrices across different layers and intermediate steps after fine-tuning the GPT2 model on the E2E dataset. Different markers represent matrices from different training steps, while different colors indicate matrices from each layer. The values of negative correlation are shown at the top, denoted by $k$.

| Partition | ModApte | ModHayes | ModLewis |
|---|---|---|---|
| 0.6 | 85.32 | 79.76 | 77.69 |
| 0.7 | 85.32 | **80.01** | **77.74** |
| 0.8 | **85.34** | 79.93 | 77.63 |
| 0.9 | 85.27 | 79.85 | 77.64 |
| 1.0 | 85.23 | 79.59 | 77.42 |

Table 18: Experiment results on different data partitions of BiDoRA

assigning different portions of data. The direction component has more trainable parameters, so it is reasonable to use more data for training the lower level while using the remaining data for training magnitudes. As shown in Table 18, We varied the inner-level dataset $\mathcal{D}_{tr}$ partition from 0.6 to 1.0 with 0.1 intervals and experimented with RoBERTa-base on three splits of the Reuters21578 dataset to examine its influence.

The results indicate that both extreme cases are negative to the overall performance. When the inner partition is too small ($\leq 0.6$), directions are not well-trained, and when the inner partition is 1.0, magnitudes are not trained at all, leading to a significant performance drop. These findings demonstrate that bi-level optimization is effective in the sense that both levels are necessary for enhancing performance. Although tuning the partition ratio may further improve overall performance, we maintain a consistent data partition of 8:2 in all the experiments for simplicity. A fixed configuration of data partition already consistently yields superior performance of BiDoRA, demonstrating that our method is robust to this hyperparameter within a certain range.

