# OpenReview forum: "BiDoRA: Bi-level Optimization-based Weight-Decomposed Low-Rank Adaptation"
_ICLR.cc/2025/Conference — Submitted to ICLR 2025_

### Official Review · Reviewer_YccA · 2024-10-31

**Soundness:** 3
**Presentation:** 3
**Contribution:** 3
**Rating:** 6
**Confidence:** 4

**Summary:**

This paper proposes BiDoRA, a novel method for parameter-efficient fine-tuning (PEFT). While based on DoRA's methodology, it introduces a bi-level optimization approach that optimizes parameters separately. The authors argue that this approach reduces overfitting and enables learning patterns more similar to full fine-tuning. Through experiments across various NLP datasets, they demonstrated that BiDoRA consistently outperforms both DoRA and other PEFT methods.

**Strengths:**

This work represents a valuable first attempt at applying bi-level optimization to PEFT, extending DoRA's magnitude/direction decomposition in a natural progression. Experimental validation includes a comprehensive evaluation across diverse NLP tasks, such as NLU, NLG, and token classification, with well-suited comparisons to baselines and detailed ablation studies demonstrating the effectiveness of each component. The implementation is both clear and reproducible, contributing to the overall technical completeness of the work.

**Weaknesses:**

Methodological Limitations: The approach may be perceived as a straightforward combination of existing methods (DoRA + bi-level optimization), which could limit its novelty. Additionally, the method introduces substantial computational complexity, running at 3.92 times the cost of LoRA, and requires extra hyperparameter tuning. To address these issues, improvements in performance relative to computational cost are needed. Developing an automated framework for setting additional hyperparameters could also streamline the process and reduce user intervention.

Experimental Limitations: The study lacks experiments involving recent large language models (LLMs), does not sufficiently analyze performance on very small datasets, and has limited exploration of data split ratio effects. To enhance experimental rigor, including evaluations with larger, contemporary models, examining performance across varying dataset sizes, and conducting sensitivity analyses across different split ratios would offer a more comprehensive assessment of the method’s effectiveness.

Theoretical Limitations: The paper currently provides limited theoretical guarantees for generalization performance and lacks convergence analysis for the bi-level optimization approach. Strengthening the theoretical foundation could be achieved by introducing an upper bound analysis for generalization error and supplying a convergence proof for the optimization algorithm.

Together, addressing these methodological, experimental, and theoretical gaps would significantly enhance the robustness and credibility of the study.

**Questions:**

1) Are there experimental results involving larger, more recent LLMs?
2) Is there an analysis for cases with very small dataset sizes?
3) Are there optimization plans in place to reduce computational complexity?
4) Is there additional analysis on how the choice of data split ratios affects performance?
5) Can theoretical guarantees for generalization error be provided?

---

> ### Author Response · Authors · 2024-11-25
>
> We appreciate your constructive feedback very much. We provide our response to your review as follows.
>
> ### Weaknesses
>
> - **Novelty**
>
>   We acknowledge that our framework does not introduce extensive novelty to the model architecture on the PEFT side or the hyper-gradient estimation method on the bi-level optimization side. However, we would like to clarify that its application of bilevel optimization in the context of PEFT, especially tailored for the DoRA method, is new and constitutes a novel contribution.
>
> - **Computational Complexity**
>
>   Although our method is more computationally costly than DoRA, it converges much faster, which compensates for the increased computational load. We provide additional results in the original computation costs section (Section 5.9). Notably, our method achieves comparable performance within the same runtime and can outperform the baselines when a longer time budget is available. This underscores the practicality of BiDoRA.
>
>   Furthermore, the efficiency of BiDoRA can be improved and computational costs reduced by using better hyper-gradient estimators such as implicit differentiation [2] and SAMA [3]. These can be easily integrated into our framework thanks to support from the Betty library [4]. Currently, we focus on the finite difference approximation scheme [1] for ease of understanding.
>
> - **Hyperparameter Tuning**
>
>   Fortunately, we found the hyperparameters relatively robust within certain ranges. While further fine-tuning could potentially improve performance, a straightforward setup already achieves superior performance compared to the baselines. Please refer to Appendix B for all hyperparameter settings. We kept the train-validation split ratio the same (train: validation=8: 2) across all experiments, and we found this ratio to be effective, though it could be fine-tuned for better performance (see Appendix F for details). We did not perform extensive tuning; a value of 0 or 1e-5 was used for all experiments.. The upper-level learning rate was set to the same value as the lower-level learning rate for all experiments without tuning.
>
>   We also appreciate your kind suggestion of using an automated framework for setting additional hyperparameters. In fact, they have been widely explored. Methods such as Bayesian optimization are available; for example, the wandb platform (https://github.com/wandb/wandb) supports automated hyperparameter tuning with minimal user intervention.
>
> - **Experiments on Larger Models and Smaller Datasets**
>
>   To address your concerns, we have improved our work by conducting two additional experiments:
>
>   - **Larger Models:** We conducted experiments with much larger-scale LLMs, including the DeBERTa-v2-xxlarge model (1.5B parameters) and the Llama2-7b model (7B parameters). The results presented in Section 5.6 indicate that BiDoRA yields better performance when fine-tuning models with a very large number of parameters.
>
>   - **Extremely Small Datasets:** We also conducted experiments with extremely small datasets and presented the results in Section 5.5. Specifically, we fine-tuned the protein language model ESM on three prediction tasks with extremely small datasets (3695, 936, and 200 training samples, respectively). In these cases, overfitting is more severe in LoRA and DoRA, and BiDoRA shows improvements with a larger margin.
>
>    The two additional results confirm our previous findings on BiDoRA's effectiveness when applied to various tasks and diverse architectures.
>
> - **Theoretical Limitations**
>
>   The convergence criteria and generalization error for our gradient-based bi-level optimization have been widely studied. Please refer to [1,2,3] and the references therein. We have also added a comment on this point at the end of Section 4.1.
>
> ### Questions
>
> Please refer to the corresponding replies to the weaknesses above.
>
> [1] Rajeswaran, Aravind, et al. "Meta-learning with implicit gradients." *Advances in neural information processing systems* 32 (2019).
>
> [2] Pedregosa, Fabian. "Hyperparameter optimization with approximate gradient." *International Conference on Machine Learning*. PMLR, 2016.
>
> [3] Bao, Fan, et al. "Stability and generalization of bilevel programming in hyperparameter optimization." *Advances in neural information processing systems* 34 (2021): 4529-4541.
>
> [4] Choe, Sang Keun, et al. "Betty: An automatic differentiation library for multilevel optimization." *arXiv preprint arXiv:2207.02849* (2022).

---

### Official Review · Reviewer_p28U · 2024-10-31

**Soundness:** 3
**Presentation:** 3
**Contribution:** 3
**Rating:** 6
**Confidence:** 3

**Summary:**

This paper proposes BiDoRA, a new fine-tuning method that:
1. Decomposes the model’s weights into two parts—magnitude and direction.
2. Optimizes each part separately: The "direction" part is trained on one dataset split (training set), while the "magnitude" part is optimized using another split (validation set).
3. Iterates between these two optimization levels to decouple the updates, helping the model generalize better.

This bi-level approach allows BiDoRA to perform more like full fine-tuning while avoiding overfitting. In experiments across various NLP tasks, BiDoRA consistently outperforms other parameter-efficient fine-tuning methods like LoRA and DoRA. The authors validate BiDoRA on natural language understanding, generation, and token classification tasks, showing that it reduces overfitting and achieves better overall performance than existing methods.

**Strengths:**

Originality: BiDoRA introduces a unique bi-level optimization approach to fine-tuning large language models (LLMs), addressing a common tradeoff in parameter-efficient fine-tuning (PEFT) between generalization and computational efficiency. By decomposing the model’s weights into magnitude and direction components and optimizing each on different data splits, BiDoRA creatively combines aspects of neural architecture search with PEFT, offering a compelling alternative to current methods like LoRA and DoRA. This methodological innovation could inspire further development in PEFT techniques across domains.

And specifically I think it's interesting that in BiDoRA, both the magnitude and direction components are trainable, but they are optimized separately through a bi-level optimization framework.

Decomposition: The model's weights are decomposed into two parts:

Magnitude (m): Represents the scale of each parameter.
Direction (V): Represents the vector defining the orientation of each weight.
Optimization Approach:
BiDoRA optimizes the direction component at the lower level using a training set split. During this phase, the magnitude remains fixed, allowing the model to focus solely on finding the optimal direction adjustments.
At the upper level, BiDoRA optimizes the magnitude component by minimizing the loss on a validation set split, with the direction component fixed. This stage uses the optimized direction from the lower level to update the magnitude through hyper gradient descent.
Iterative Training:

The magnitude and direction components are iteratively updated until convergence. This iterative decoupling enables each component to learn independently while promoting generalization and reducing overfitting.
This bi-level approach allows each component to be optimized in a way that would ideally yield better performance than traditional PEFT methods, which tend to optimize these components together, leading to coupled gradients and potentially reduced flexibility.

Quality: The paper demonstrates a high level of rigor, with comprehensive experimental evaluations across multiple tasks, including natural language understanding, generation, and token classification. The experiments span diverse datasets, and the authors include robust analyses, such as ablation studies and weight decomposition, that substantiate the effectiveness of the bi-level optimization framework. This thorough evaluation establishes BiDoRA’s consistent performance advantage over existing PEFT methods, solidifying the validity of the approach.

Clarity: The paper is well-organized and clearly articulates both the theoretical underpinnings and practical implications of BiDoRA. Key concepts, such as weight decomposition and bi-level optimization, are introduced methodically, and diagrams aid in visualizing the process. Algorithmic details, hyperparameters, and dataset splits are documented with transparency, making it easier for readers and future researchers to reproduce and build upon this work.

Significance: The introduction of BiDoRA is a valuable contribution to the field of large language model fine-tuning, where computational efficiency is increasingly critical. By reducing overfitting and enhancing generalization, BiDoRA effectively bridges the gap between parameter efficiency and performance, providing a practical solution that could impact a wide range of applications requiring adaptable, high-performing LLMs. Its relevance to real-world LLM deployment in resource-constrained settings adds to the significance of its contributions.

**Weaknesses:**

Computational Cost and Efficiency: Although BiDoRA shows a significant performance improvement, the bi-level optimization approach introduces a high computational cost, as reported with nearly fourfold overhead compared to LoRA. This could limit BiDoRA’s practicality in scenarios where resources are constrained. An in-depth analysis of ways to reduce computational complexity without sacrificing performance—such as approximations, alternative regularization techniques, or a comparative exploration of optimization strategies with fewer computational steps—would enhance the method’s accessibility.

Hyperparameter Sensitivity and Tuning: The paper provides limited insight into the sensitivity of BiDoRA to hyperparameters like data split ratios, learning rates for each level, and the orthogonal regularization coefficient. Given that bi-level optimization frameworks often require precise tuning, a sensitivity analysis would help clarify the stability of BiDoRA’s performance under various configurations. This analysis would also support practitioners in understanding which parameters are most influential, guiding them in practical applications.

Scope of Evaluation on Model Types and Tasks: While BiDoRA’s evaluation spans diverse NLP tasks, it primarily focuses on RoBERTa and GPT-2 models. Expanding the scope to include additional architectures, particularly those with different structural properties (e.g., BERT or T5), would verify BiDoRA’s generalizability across model types. Furthermore, testing BiDoRA in non-NLP tasks, such as vision or multimodal tasks, would strengthen the claim of BiDoRA as a general-purpose PEFT framework and broaden its potential applications.

Limited Exploration of Overfitting Mitigation: The paper emphasizes BiDoRA’s success in mitigating overfitting through the decoupling of magnitude and direction optimizations. However, it does not provide an in-depth comparison with existing regularization techniques or alternative overfitting mitigation strategies within PEFT. A comparison or ablation study exploring how BiDoRA’s approach to overfitting compares with alternative techniques (e.g., dropout or adaptive regularization) would better demonstrate the unique value of the bi-level approach.

**Questions:**

Could the authors provide additional details on the computational efficiency of BiDoRA? Given the reported computational overhead (approximately four times that of LoRA), are there strategies under consideration for reducing this cost? For instance, would approximate or adaptive bi-level optimization approaches be feasible to explore?

Since bi-level optimization can sometimes suffer from convergence issues, could the authors clarify the convergence criteria used in BiDoRA? Additionally, are there any stability challenges encountered during training, and if so, how were these addressed? This information would be valuable for practitioners looking to implement BiDoRA in different contexts.

---

> ### Author Response · Authors · 2024-11-25
>
> We appreciate your constructive feedback very much. We provide our response to your review as follows.
>
> ### Weaknesses
>
> - **Computational Cost and Efficiency**
>
>   We would like to clarify that although our method is computationally costly than DoRA, it converges much faster, as shown by our additional results in Figure 4 (Section 5.9). It helps mitigate the computational overhead to some extent and allows BiDoRA to achieve comparable performance at the same wall-clock time. Moreover, BiDoRA can outperform the baselines with a longer time budget, underscoring its practicality.
>
> - **Hyperparameter Sensitivity and Tuning**
>
>   Fortunately, we found the hyperparameters relatively robust within certain ranges. While further fine-tuning could potentially improve performance, a straightforward setup already achieves superior performance compared to the baselines. Please refer to Appendix B for all hyperparameter settings. We kept the train-validation split ratio the same (train: validation=8: 2) across all experiments, and we found this ratio to be effective, though it could be fine-tuned for better performance (see Appendix F for details). We did not perform extensive tuning; a value of 0 or 1e-5 was used for all experiments. The upper-level learning rate was set to the same value as the lower-level learning rate for all experiments without tuning.
>
> - **Scope of Evaluation on Model Types and Tasks**
>
>   To address your concerns, we have improved our work by conducting two additional experiments:
>
>   - **Larger Models:** We conducted experiments with much larger-scale LLMs, including the DeBERTa-v2-xxlarge model (1.5B parameters) and the Llama2-7b model (7B parameters). The results presented in Section 5.6 indicate that BiDoRA yields better performance when fine-tuning models with a very large number of parameters.
>   - **Extremely Small Datasets:** We also conducted experiments with extremely small datasets and presented the results in Section 5.5. Specifically, we fine-tuned the protein language model ESM on three prediction tasks with extremely small datasets (3695, 936, and 200 training samples, respectively). In these cases, overfitting is more severe in LoRA and DoRA, and BiDoRA shows improvements with a larger margin.
>
>   The encouraging results suggest that BiDoRA can serve as a general-purpose PEFT framework.
>
> - **Limited Exploration of Overfitting Mitigation**
>
>   We have indeed considered other potential general-purpose techniques for alleviating overfitting, such as weight decay and dropout. Specifically, we have already tuned the weight decay coefficient and dropout ratio to optimize performance for the baselines. However, we found that while having applied these techniques, they were insufficient for DoRA, particularly. On the other hand, our framework leverages the characteristics of DoRA, which further addresses the overfitting issue.
>
> ### Questions
>
> - **Improving Computational Efficiency**
>
>   In fact, the most computationally expensive part of our method is the hyper-gradient estimation. Therefore, the efficiency of BiDoRA could be improved by using more accurate hyper-gradient estimators, such as implicit differentiation [2] and SAMA [3]. These can be easily integrated into our framework thanks to support from the Betty library [4]. We leave this potential solution to future work and have added a comment on this at the end of Section 6. Currently, we focus on the finite difference approximation scheme [1] for ease of understanding.
>
> - **Convergence Criteria**
>
>   The convergence criteria for our gradient-based bi-level optimization are well-established. Please refer to [2,5] and the references therein. We have also added a comment on this point at the end of Section 4.1.
>
> [1] Liu, Hanxiao, Karen Simonyan, and Yiming Yang. "Darts: Differentiable architecture search." *arXiv preprint arXiv:1806.09055* (2018).
>
> [2] Rajeswaran, Aravind, et al. "Meta-learning with implicit gradients." *Advances in neural information processing systems* 32 (2019).
>
> [3] Choe, Sang, et al. "Making scalable meta learning practical." *Advances in neural information processing systems* 36 (2024).
>
> [4] Choe, Sang Keun, et al. "Betty: An automatic differentiation library for multilevel optimization." *arXiv preprint arXiv:2207.02849* (2022).
>
> [5] Pedregosa, Fabian. "Hyperparameter optimization with approximate gradient." *International conference on machine learning*. PMLR, 2016.

---

> > ### Comment · Reviewer_p28U · 2024-11-25
> > **Response to authors' comments**
> >
> > Thank you for your detailed and thoughtful responses to my feedback. I appreciate the effort you have put into addressing the concerns raised in my review. Below, I provide my follow-up thoughts on your clarifications and additional experiments:
> >
> > 1. Computational Cost and Efficiency
> > I acknowledge your clarification regarding BiDoRA’s faster convergence and comparable wall-clock performance to baseline methods like LoRA. The additional results in Figure 4 are helpful in understanding the trade-offs. Highlighting this in the main text of the paper, rather than relying on a single section, could strengthen the practical relevance of BiDoRA for readers evaluating its feasibility.
> >
> > The suggestion to improve computational efficiency using techniques like implicit differentiation and SAMA is promising. Including a brief mention of these methods as future work in the conclusion section was a good move. These insights make your framework more actionable for researchers interested in computational optimization.
> >
> > 2. Hyperparameter Sensitivity and Tuning
> > Your explanation regarding hyperparameter robustness is reassuring. The information in Appendices B and F provides valuable context. However, including a condensed version of this robustness analysis in the main text (e.g., Section 4 or 5) could emphasize BiDoRA’s stability, which is critical for practitioners. I also appreciate the consistency in your choice of split ratios and learning rates. Exploring the possibility of an adaptive train-validation split ratio as future work might further enhance the method's practicality.
> >
> > 3. Scope of Evaluation on Model Types and Tasks
> > The inclusion of additional experiments with larger models (DeBERTa-v2-xxlarge, Llama2-7b) and extremely small datasets significantly strengthens the evaluation scope. These results in Sections 5.5 and 5.6 effectively demonstrate BiDoRA’s versatility across different parameter scales and dataset sizes.
> >
> > The protein language model experiments are particularly noteworthy, as they highlight BiDoRA’s capacity to handle overfitting in low-data regimes. Expanding upon these findings in the main text to elaborate on BiDoRA’s potential applications beyond NLP could further enhance its impact.
> >
> > 4. Limited Exploration of Overfitting Mitigation
> > Your explanation regarding the integration of existing regularization techniques (e.g., weight decay, dropout) and their relative effectiveness compared to BiDoRA is well-received. The emphasis on leveraging DoRA’s characteristics to address overfitting is a compelling argument. A comparative ablation study focusing specifically on these regularization techniques versus BiDoRA could provide additional quantitative support for this claim.

---

> > > ### Author Response · Authors · 2024-11-26
> > >
> > > Dear Reviewer,
> > >
> > > Thank you for your response and kind reminder about more ablation experiments and revising the main text. We are working on this, and we will start posting it as soon as possible.
> > >
> > > Thank you again for raising this.
> > >
> > > Best regards,
> > > Authors

---

### Official Review · Reviewer_MKru · 2024-11-08

**Soundness:** 1
**Presentation:** 3
**Contribution:** 1
**Rating:** 3
**Confidence:** 5

**Summary:**

Low-Rank Adaptation (LoRA) is a parameter-efficient fine-tuning (PEFT) method. Weighted Decomposed Low-Rank Adaptation (DoRA) is a variant of LoRA, where the update matrix is decomposed into two components: magnitude and direction.

This paper argues that the additional parameters in DoRA increase the risk of overfitting, and simultaneously optimizing both magnitude and direction limits its learning capacity. To address these issues, the authors introduce Bi-Level Optimization-Based Weight-Decomposed Low-Rank Adaptation (BiDoRA), which integrates bi-level optimization into DoRA.

The core idea of BiDoRA is to separate the optimization of magnitude and direction. In the first phase, BiDoRA alternately trains magnitude and direction until convergence. Additionally, the training data for these two components is separated to further prevent overfitting: the magnitude is trained on a validation set, while the direction is trained on a training set. In the second phase, the direction is further trained on the combined dataset until convergence.

Experiments are conducted on natural language understanding (NLU), natural language generation (NLG), and token classification tasks. BiDoRA shows marginal improvements over the baselines, with almost all gains being less than 1 percentage point. Ablation studies reveal that the effects of different design components are relatively small. The training time for BiDoRA is 3.92 times that of LoRA, while DoRA requires 1.3 times the training time of LoRA.

**Strengths:**

* The paper is well-written and easy to understand.
* The authors' efforts to evaluate across multiple tasks are commendable. However, there are still many problems with these evaluations (see Weaknesses).

**Weaknesses:**

1. **The motivation of this paper appears to be questionable.** The authors claim that DoRA increases the risk of overfitting, basing this on two pieces of evidence:

   - DoRA introduces additional parameters compared to LoRA.
   - The gap between training and test accuracy curves for DoRA is larger than that of BiDoRA.

   However, these two points do not convincingly support the claim. First, while additional parameters can sometimes contribute to overfitting, they are not a sufficient condition for it. In fact, DoRA adds only a negligible number of parameters (0.01% of the model size, as reported by the authors) beyond LoRA. Moreover, prior work [1] suggests that LoRA learns less than full fine-tuning and may even act as a form of regularization, implying that the risk of overfitting is generally low across these PEFT methods.

   Additionally, the training curves are not necessarily indicative of overfitting, as they can be significantly influenced by factors such as hyperparameters, model architecture, and dataset characteristics. The authors present results from only a single configuration, which limits the generalizability of their findings.

   Finally, the authors’ attribution of an *alleged overfitting problem* to DoRA’s concurrent training lacks a strong foundation.

2. **The proposed BiDoRA method is overly complex and difficult to use.** It requires a two-phase training process, with the first phase itself consisting of two sub-steps. It also introduces two additional hyperparameters: the weight of orthogonality regularization and a ratio for splitting training and validation sets. As a result, BiDoRA takes 3.92 times longer to train than LoRA.

3. **Performance differences between methods are minimal across evaluations**. In nearly all results, the performance differences between the methods are less than 1 percentage point, which may be attributable to random variation. Furthermore, the benchmarks selected are outdated and likely saturated.

[1] [LoRA Learns Less and Forgets Less](https://arxiv.org/abs/2405.09673)

**Questions:**

The results in Table 3 do not align with those reported in the original LoRA paper. In the original paper, LoRA achieves a BLEU score of 70.4 with 0.35M parameters. However, in your results, LoRA only reaches 63.7 BLEU with 0.39M parameters. This discrepancy suggests there may be errors in your evaluation or setup.

---

> ### Author Response · Authors · 2024-11-25
>
> We appreciate your constructive feedback very much. We provide our response to your review as follows.
>
> ### Weaknesses
>
> - **Weakness 1**
>
>   Thank you for raising this point. We would like to clarify that although LoRA learns less and can behave as regularization, it is still insufficient when data is very limited. Overfitting in PEFT has been observed in existing work, especially with very limited data [1]. In such cases, the fine-tuned part remains over-parameterized, with the number of parameters (> 100k) much larger than the number of samples (in the hundreds). Additionally, although DoRA introduces fewer parameters than LoRA, the number is still significant when scaling up. For models around 10B parameters, the 0.01% increment will still result in a significant parameter increment, being much larger than the size of many downstream tasks (for example, some tasks in the GLUE benchmark, like CoLA, RTE, MRPC, and STS-B, have sizes in the magnitude of 10k). The overfitting issues become more severe when conducting evaluation on extremely small datasets, where BiDoRA achieves more performance improvements. The additional results are presented in Section 5.6 of our rebuttal version.
>
> - **Weakness 2**
>
>   - Implementation
>
>     The Betty framework [2] already provides an efficient implementation, so no significant effort was required to develop our algorithm.
>
>   - Hyperparameters
>
>     Fortunately, we found the hyperparameters relatively robust within certain ranges. While further fine-tuning could potentially improve performance, a straightforward setup already achieves superior performance compared to the baselines. Please refer to Appendix B for all hyperparameter settings. We kept the train-validation split ratio the same (train: validation=8: 2) across all experiments, and we found this ratio to be effective, though it could be fine-tuned for better performance (see Appendix F for details). We did not perform extensive tuning; a value of 0 or 1e-5 was used for all experiments. The upper-level learning rate was set to the same value as the lower-level learning rate for all experiments without tuning.
>
>   - Computational Cost
>
>     We would like to clarify that although our method is computationally costly than DoRA, it converges much faster, as shown by our additional results in Figure 4 (Section 5.9). It helps mitigate the computational overhead to some extent and allows BiDoRA to achieve comparable performance at the same wall-clock time. Moreover, BiDoRA can outperform the baselines with a longer time budget, underscoring its practicality.
>
> - **Weakness 3**
>
>   The GLUE benchmark has been widely used and may be saturated, which may explain the marginal performance improvement. However, we evaluated all methods over five runs on different seeds and reported the average. The standard deviation is also small (comparable to LoRA, less than 0.3 for most cases in the GLUE benchmark). Considering BiDoRA consistently outperforms DoRA, it is not likely due to random variation.
>
>   To address your concerns, we have improved our work by conducting two additional experiments:
>
>   - **Larger Models:** We conducted experiments with much larger-scale LLMs, including the DeBERTa-v2-xxlarge model (1.5B parameters) and the Llama2-7b model (7B parameters). The results presented in Section 5.6 indicate that BiDoRA yields better performance when fine-tuning models with a very large number of parameters.
>
>   - **Extremely Small Datasets:** We also conducted experiments with extremely small datasets and presented the results in Section 5.5. Specifically, we fine-tuned the protein language model ESM on three prediction tasks with extremely small datasets (3695, 936, and 200 training samples, respectively). In these cases, overfitting is more severe in LoRA and DoRA, and BiDoRA shows improvements with a larger margin.
>
>    The two additional results confirm our previous findings on BiDoRA's effectiveness when applied to various tasks and diverse architectures.
>
> ### Questions
>
> - **Question 1**
>
>   We apologize for the typo in the table; the actual performance of LoRA has been updated in Table 3. Due to computational resource limitations, we reevaluated all baselines with fewer but the same total iterations, which may have resulted in slight performance degradation for all compared methods. Despite this, we ensured that the comparison settings were the same for all methods to maintain fairness.
>
>
>
> [1] Zhang, Li, et al. "BLO-SAM: Bi-level Optimization Based Finetuning of the Segment Anything Model for Overfitting-Preventing Semantic Segmentation." *Forty-first International Conference on Machine Learning*.
>
> [2] Choe, Sang Keun, et al. "Betty: An automatic differentiation library for multilevel optimization." *arXiv preprint arXiv:2207.02849* (2022).

---

> > ### Comment · Reviewer_MKru · 2024-11-26
> >
> > Thank you for your response. It has alleviated some of my concerns, but overall, I maintain my original evaluation. Below is my updated review.
> >
> > * **On the Motivation**
> >
> >   * You clarified that the overfitting problem you aim to address arises in scenarios with extremely limited data, rather than in general cases. I think it’s good to define the scope of your work clearly. However, this emphasis is insufficient in the paper, particularly in the abstract and introduction, where it should be highlighted to adjust readers' expectations about your actual contributions.
> >   * Despite this, I still believe the motivation is unconvincing:
> >     - **You argue that more parameters lead to overfitting, but this claim contradicts your experimental results.** In Tables 3, 5, 6, and 7, Full FT has parameter counts hundreds of times larger than LoRA and DoRA, yet its generalization performance is generally better. This contradiction is particularly evident in Tables 5, 6, and 7, which present new experiments on extremely small datasets. These results significantly undermine your assertion. If Full FT with hundreds of times more parameters than LoRA does not exhibit significant overfitting in such scenarios, your claim that the additional 0.01% parameters in DoRA would cause overfitting is even less tenable.
> >     - Even assuming that large parameter counts cause overfitting, **there are many more practical solutions available.** For instance, one could avoid using an overly large base model (why use a very large model if overfitting is a concern?), reduce the rank of LoRA, or apply LoRA to only specific modules or layers. Not to mention other methods like dropout, weight decay, etc., which are mature, simple, widely validated, and well-studied in the community. I see no evidence that your proposed method offers advantages over these alternatives.
> >
> > * **On the Complexity of the Method**
> >
> >   This is unrelated to which framework implements your method. I am referring to the intrinsic complexity of your method itself, including the intricate pipeline, additional hyperparameters, and a new loss term.

---

> > > ### Author Response · Authors · 2024-11-28
> > >
> > > Dear reviewer,
> > >
> > > We appreciate your time and detailed feedback, which will be valuable for improving our work. We will revise the paper according to your suggestions to improve its quality, especially by clarifying our motivation and reducing its complexity. Thank you for your efforts in making this research area better through your thoughtful reviews.
> > >
> > > Best, Authors

---

### Official Review · Reviewer_wcvc · 2024-11-09

**Soundness:** 2
**Presentation:** 3
**Contribution:** 2
**Rating:** 3
**Confidence:** 3

**Summary:**

This paper proposes a method for parameter efficient fine tuning using bi-level optimization that competes with LoRA. Results test it on training RoBERTa and GPT-2 medium for NLU and NLG tasks respectively. The overall results demonstrate consistently better results than LoRA and DoRA.

**Strengths:**

- In experiments, the proposed method demonstrates consistently superior results compared to the others.
- The paper was written reasonably clearly, so I was able to follow the equations.

**Weaknesses:**

- The motivation for the bi-level optimization approach was a bit vague. I did read the explanation in the introduction, but it says "Furthermore, in DoRA, the magnitude and incremental direction components are optimized concurrently, leading to a highly constrained updating pattern that may overlook the diverse learning patterns required for different downstream tasks." It is not clear what this means concretely...
- Experiments were on models that do not represent the state of the art at this point. It would be much more convincing if the methods were demonstrated to work on more powerful models (e.g. at least 7B).
- As noted in 5.7, the proposed method is accompanied with significant additional training time, which may limit its practical usefulness.

**Questions:**

None

---

> ### Author Response · Authors · 2024-11-25
>
> We appreciate your constructive feedback very much. We provide our response to your review as follows.
>
> ### Weaknesses
>
> - **Weakness 1**
>
>   We apologize for not making this point clear. This motivation is based on the empirical findings of DoRA [1]. In DoRA, the authors found that the magnitude and direction components (defined in their weight decomposition analysis) are highly coupled in the original LoRA method. This coupling may limit LoRA's learning capacity. To address this, they decoupled LoRA by implementing an explicit reparameterization. Building on the same motivation, we aim to further decouple the two components by enabling them to be optimized separately in two distinct levels and data splits, providing more flexibility and thus achieving higher learning capacity than both LoRA and DoRA.
>
> - **Weakness 2**
>
>   To address your concerns, we have improved our work by conducting experiments on larger models, including the DeBERTa-v2-xxlarge model (1.5B parameters) and the Llama2-7b model (7B parameters). The results presented in Section 5.6 indicate that BiDoRA yields better performance when fine-tuning models with a very large number of parameters.
>
> - **Weakness 3**
>
>   We would like to clarify that although our method is computationally costly than DoRA, it converges much faster, as shown by our additional results in Figure 4 (Section 5.9). It helps mitigate the computational overhead to some extent, and allows BiDoRA to achieve comparable performance in the same wall-clock time. Moreover, BiDoRA can outperform the baselines with a longer time budget, underscoring its practicality.
>
> [1] Liu, Shih-Yang, et al. "Dora: Weight-decomposed low-rank adaptation." *arXiv preprint arXiv:2402.09353* (2024).

---

### Author Response · Authors · 2024-11-25
**General Response**

Dear reviewers,

We have submitted the rebuttal version of the paper with major changes highlighted in purple. We highly value your concerns regarding insufficient experiments, computation efficiency, and hyperparameter sensitivity. We provide the following response in the hope of addressing your concerns.

### Further Experimental Studies

We thank all reviewers for raising concerns about the lack of diverse experimental studies. To address your concerns, we have improved our work by conducting two additional experiments:

- **Larger Models:** We conducted experiments with much larger-scale LLMs, including the DeBERTa-v2-xxlarge model (1.5B parameters) and the Llama2-7b model (7B parameters). The results presented in Section 5.6 indicate that BiDoRA yields better performance when fine-tuning models with a very large number of parameters.

- **Extremely Small Datasets:** We also conducted experiments with extremely small datasets and presented the results in Section 5.5. Specifically, we fine-tuned the protein language model ESM on three prediction tasks with extremely small datasets (3695, 936, and 200 training samples, respectively). In these cases, overfitting is more severe in LoRA and DoRA, and BiDoRA shows improvements with a larger margin.

 The two additional results confirm our previous findings on BiDoRA's effectiveness when applied to various tasks and diverse architectures.

### Computational Efficiency

We thank all reviewers for raising concerns about computational efficiency. We would like to clarify that although our method is computationally costly than DoRA, it converges much faster, as shown by our additional results in Figure 4 (Section 5.9). It helps mitigate the computational overhead to some extent, and allows BiDoRA to achieve comparable performance in the same wall-clock time. Moreover, BiDoRA can outperform the baselines with a longer time budget, underscoring its practicality.

In fact, the most computationally expensive part of our method is the hyper-gradient estimation. Therefore, the efficiency of BiDoRA could be improved by using more accurate hyper-gradient estimators, such as implicit differentiation [2] and SAMA [3]. These can be easily integrated into our framework thanks to support from the Betty library [4]. We leave this potential solution to future work and have added a comment on this at the end of Section 6. Currently, we focus on the finite difference approximation scheme [1] for ease of understanding.

### Hyperparameter Sensitivity

We are aware that BiDoRA introduces additional hyperparameters such as dataset split ratio, upper-level learning rate, and regularization coefficient. Fortunately, we found them relatively robust within certain ranges. While further fine-tuning could potentially improve performance, a straightforward setup already achieves superior performance compared to the baselines. Please refer to Appendix B for all hyperparameter settings. Specifically, we set the additional parameters in the following ways:

- **Train-validation Split Ratio:** We kept the train-validation split ratio the same (train: validation=8: 2) across all experiments, and we found this ratio to be effective, though it could be fine-tuned for better performance (see Appendix F for details).
- **Regularization Coefficient:** We did not perform extensive tuning; a value of 0 or 1e-5 was used for all experiments.
- **Upper-level Learning Rate:** Is was set to the same value as the lower-level learning rate for all experiments without tuning.

In conclusion, our method achieves superior performance without extensive hyperparameter tuning, making BiDoRA practical and easy to use.



We are open to further discussions regarding the rebuttal version. If you have any additional questions, please feel free to share them. Again, thank you all for your valuable reviews of our work.

[1] Liu, Hanxiao, Karen Simonyan, and Yiming Yang. "Darts: Differentiable architecture search." *arXiv preprint arXiv:1806.09055* (2018).

[2] Rajeswaran, Aravind, et al. "Meta-learning with implicit gradients." *Advances in neural information processing systems* 32 (2019).

[3] Choe, Sang, et al. "Making scalable meta learning practical." *Advances in neural information processing systems* 36 (2024).

[4] Choe, Sang Keun, et al. "Betty: An automatic differentiation library for multilevel optimization." *arXiv preprint arXiv:2207.02849* (2022).

---

### Meta-Review · Area_Chair_5Hb9 · 2024-12-14

**Metareview:**

BiDoRA is a bi-level optimization method designed for parameter-efficient fine-tuning (PEFT) of large language models (LLMs). It builds on DoRA by decoupling the optimization of magnitude and direction components, offering a novel perspective on fine-tuning. The paper is well-written and accessible.

However, all reviewers noted that the significant additional training time could limit its practical utility, despite its ability to achieve fast convergence. Furthermore, the evaluation results show only minimal improvements on the GLUE benchmark.

I believe the paper would benefit from another round of review, focusing on refining its motivation, presenting more robust experiments, providing stronger theoretical support, and conducting an in-depth comparison with existing regularization techniques or alternative strategies for mitigating overfitting.

Therefore, I recommend rejection.

**Additional Comments On Reviewer Discussion:**

In summary, the reviewers' concerns primarily focus on the paper's motivation, usage, computational cost, and evaluation results. While the authors have provided responses to these issues, they do not fully address the reviewers' confusion, particularly that of reviewer MKru. As a result, I support the reviewers' recommendation to reject this paper.

---

### Decision · Program_Chairs · 2025-01-22

Reject